# Cutaneous and acral melanoma cross-OMICs reveals prognostic cancer drivers associated with pathobiology and ultraviolet exposure

Anna Luiza Silva Almeida Vicente [1,2✉], Alexei Novoloaca[2], Vincent Cahais[2], Zainab Awada[2], Cyrille Cuenin[2], Natália Spitz [2], André Lopes Carvalho[1,3], Adriane Feijó Evangelista [1], Camila Souza Crovador[1], Rui Manuel Reis [1,4,5], Zdenko Herceg [2,7], Vinicius de Lima Vazquez [1,6,7] & Akram Ghantous [2,7✉]

Ultraviolet radiation (UV) is causally linked to cutaneous melanoma, yet the underlying epigenetic mechanisms, known as molecular sensors of exposure, have not been characterized in clinical biospecimens. Here, we integrate clinical, epigenome (DNA methylome), genome and transcriptome profiling of 112 cutaneous melanoma from two multi-ethnic cohorts. We identify UV-related alterations in regulatory regions and immunological pathways, with multi-OMICs cancer driver potential affecting patient survival. *TAPBP*, the top gene, is critically involved in immune function and encompasses several UV-altered methylation sites that were validated by targeted sequencing, providing cost-effective opportunities for clinical application. The DNA methylome also reveals non UV-related aberrations underlying pathological differences between the cutaneous and 17 acral melanomas. Unsupervised epigenomic mapping demonstrated that non UV-mutant cutaneous melanoma more closely resembles acral rather than UV-exposed cutaneous melanoma, with the latter showing better patient prognosis than the other two forms. These gene-environment interactions reveal translationally impactful mechanisms in melanomagenesis.

---

[1] Molecular Oncology Research Center, Barretos Cancer Hospital, Barretos, São Paulo, Brazil. [2] Epigenomics and Mechanisms Branch, International Agency for Research on Cancer (IARC), Lyon, France. [3] Early Detection Prevention and Infections Branch, International Agency for Research on Cancer (IARC), Lyon, France. [4] Life and Health Sciences Research Institute (ICVS), Medical School, University of Minho, Braga, Portugal. [5] ICVS/3B's-PT Government Associate Laboratory, Braga, Guimarães, Portugal. [6] Department of Surgery—Melanoma and Sarcoma, Barretos Cancer Hospital, Barretos, São Paulo, Brazil. [7] These authors contributed equally: Zdenko Herceg, Vinicius de Lima Vazquez, Akram Ghantous. ✉email: annaluizaalmeida@hotmail.com; GhantousA@iarc.who.int

Melanoma is a neoplasm arising from melanocytes in the skin, mucosa, or uvea[1]. It accounts for more than 75% of skin cancer-related deaths though it represents <5% of all cutaneous malignancies[2]. The incidence of melanoma has been increasing worldwide[3] and this trend has been observed for decades in some populations (e.g., the US)[4].

Epidemiological studies have highlighted that the strongest risk factors for cutaneous melanoma development are severe sunburns during childhood and intense intermittent ultraviolet (UV) exposure, which consists of UVC (100–280 nm), UVB (290–320 nm), and UVA (320–400 nm)[5]. However, there are types of melanoma that arise in body parts protected from direct UV light, and these are acral, mucosal and uveal melanomas. These types represent uncommon cancers, among which the most frequent is the acral melanoma, which occurs on the glabrous skin (the skin of palms of the hands and the soles of the feet) and the subungual area[6,7]. Even though it is rare in the general population, acral melanoma is the most common melanoma among people with darker skin[8].

The melanoma genome has the highest mutation burden of any cancer and a predominant C>T nucleotide transition signature attributable to UV radiation[9,10]. Recently, ten mutated UV signature genes were identified in both clinical samples and animal models, and patients harboring the UV mutation signature presented longer disease-free and overall survival[11]. Although associations between genetic changes and UV exposure have been well characterized, the role of epigenetic modifications induced by UV exposure has never been investigated directly in human melanoma tissues (Supplementary Data 1). Epigenetic mechanisms function as central players in tumorigenesis and as molecular sensors to environmental factors[12]. In fact, CpG DNA methylation sites are highly sensitive to UV damage, as evidenced from experimental approaches of UV exposure using cell line and animal models[13].

Furthermore, the DNA methylation profile of acral melanomas is barely characterized, which could be due in part to its scarcity. It is also unclear whether molecular differences between UV-related and non UV-related melanoma types are due to intrinsic pathological characteristics, extrinsic responses to UV exposure or a combination of both. To address these questions, a comparative study encompassing both cutaneous and acral melanomas would represent an important step forward, with particular focus on epigenetic mechanisms as they can function as both sensors to exposures and key determinants of cell identity. The most recent melanoma classification by the World Health Organization (WHO), including the Blue Books by the International Agency for Research on Cancer (IARC), presented evidence based on epidemiologic, clinical, histopathologic and genomic features[14], while not yet encompassing epigenomics.

We hypothesize that epigenetic alterations, interplaying with transcriptional and mutational events, constitute critical biological mechanisms underpinning intrinsic pathological differences and extrinsic responses to UV exposure in cutaneous and acral melanomas. We perform differential DNA methylome-wide analysis in cutaneous melanoma patients comparing UV-exposed and non UV-exposed melanomas in two independent clinical cohorts, including a sample population from Brazil which encompasses the white and pigmented phenotypes (Fig. 1). UV exposure status is inferred from UV mutational signatures derived from whole genome sequencing (WGS) or whole exome sequencing (WES). This is followed by functional genomic, pathway and methylation-expression analysis of the identified DNA methylation alterations, assessment of their cancer driver roles using a multi-OMICs approach, investigation of their effect on patient survival, and validation of the top hits using bisulfite pyrosequencing. The methylome landscape of cutaneous melanoma is then compared to that of acral melanoma to elucidate the relative contributions of intrinsic pathological and extrinsic UV-related differences towards shaping the cancer epigenome of the two major UV-related and non UV-related melanoma types (Fig. 1).

## Results

### Cross genome-methylome analysis of UV exposure in cutaneous melanoma.
UV mutation status was inferred in cutaneous melanoma patients using WGS and WES data from Barretos Cancer Hospital (BCH) in the context of International Cancer Genome Consortium (ICGC)-Brazil project and The Cancer Genome Atlas (TCGA) study, respectively (Fig. 1). Similar characteristics were observed in the BCH and TCGA-cutaneous melanoma patients, including larger proportions of the male sex, white skin phenotype, metastatic tumor type, UV mutation signature, and BRAF molecular group (Table 1). Primary tumors and *BRAF* mutations were relatively more enriched in BCH than in TCGA patients (Table 1, $p = 9.40e-03$, $p = 3.20e-03$, respectively).

We observed that UV-mutant cutaneous melanoma patients have higher melanoma-specific survival relative to non UV-mutant patients in both BCH and TCGA (Fig. 2a). In order to investigate whether the DNA methylome functions as a molecular sensor to UV exposure and underlies the difference in survival between UV-mutant *versus* non UV-mutant cutaneous melanoma patients, DNA methylome-wide analysis based on Infinium HumanMethylation450 (450 K) array was performed in BCH samples and compared with that in the TCGA cohort (the quality-control analysis and selection of the appropriate statistical model, including adjustment for potential confounders, are described in the Methods section and Supplementary Data 2–5).

In BCH melanomas, of the 2620 differentially methylated regions (DMRs), 1541 (58.8%) were hypermethylated and 1,079 (41.2%) were hypomethylated (Fig. 2b; Supplementary Data 2). A similar proportion of hypermethylated (62.8%; 378 out of 602) and hypomethylated (37.2%; 224 out of 602) DMRs was observed in TCGA (Fig. 2b; Supplementary Data 6). The enrichment distributions in CpG regulatory or density regions were also similar in both cohorts. Specifically, in CpG regulatory regions, the significant enrichments in both cohorts were those of hypomethylated DMRs in regions 1–5 Kb upstream of the transcription start site and of hypermethylated DMRs in promoters, exon/intron boundaries and 5′UTR ($p < 0.001$) (Fig. 2c). In CpG density regions, the significant enrichments were those of hyper- or hypomethylated DMRs in CpG islands or shores ($p < 0.001$) (Fig. 2d).

### The DNA methylome marks UV exposure with effect on immunomodulation.
In order to prioritize the top DMRs that distinguish UV-mutant and non UV-mutant cutaneous melanoma patients (Supplementary Data 7, 8), we applied the filters described in Supplementary Fig. 1a to focus on DMRs encompassing at least 3 CpGs, with consistent directions of effect, absolute effect sizes ≥10%, and not enriched in single-nucleotide polymorphisms (SNPs). The resultant methylome map distinctly clustered UV-mutant from non UV-mutant patients in both BCH and TCGA (Fig. 3a). In the BCH cohort, cluster C1 (as defined by Euclidean distance) was fully occupied by non UV-mutant samples (Fig. 3a) and exhibited a DNA methylation profile that was visually distinct, with an upper hypermethylation (red) stretch and a lower hypomethylation (blue) stretch, relative to the other clusters. Even if C1 is merged with the adjacent cluster C2, the non UV-mutant patients remain statistically enriched in this combined cluster ($p = 1.95e-03$), which now encompasses

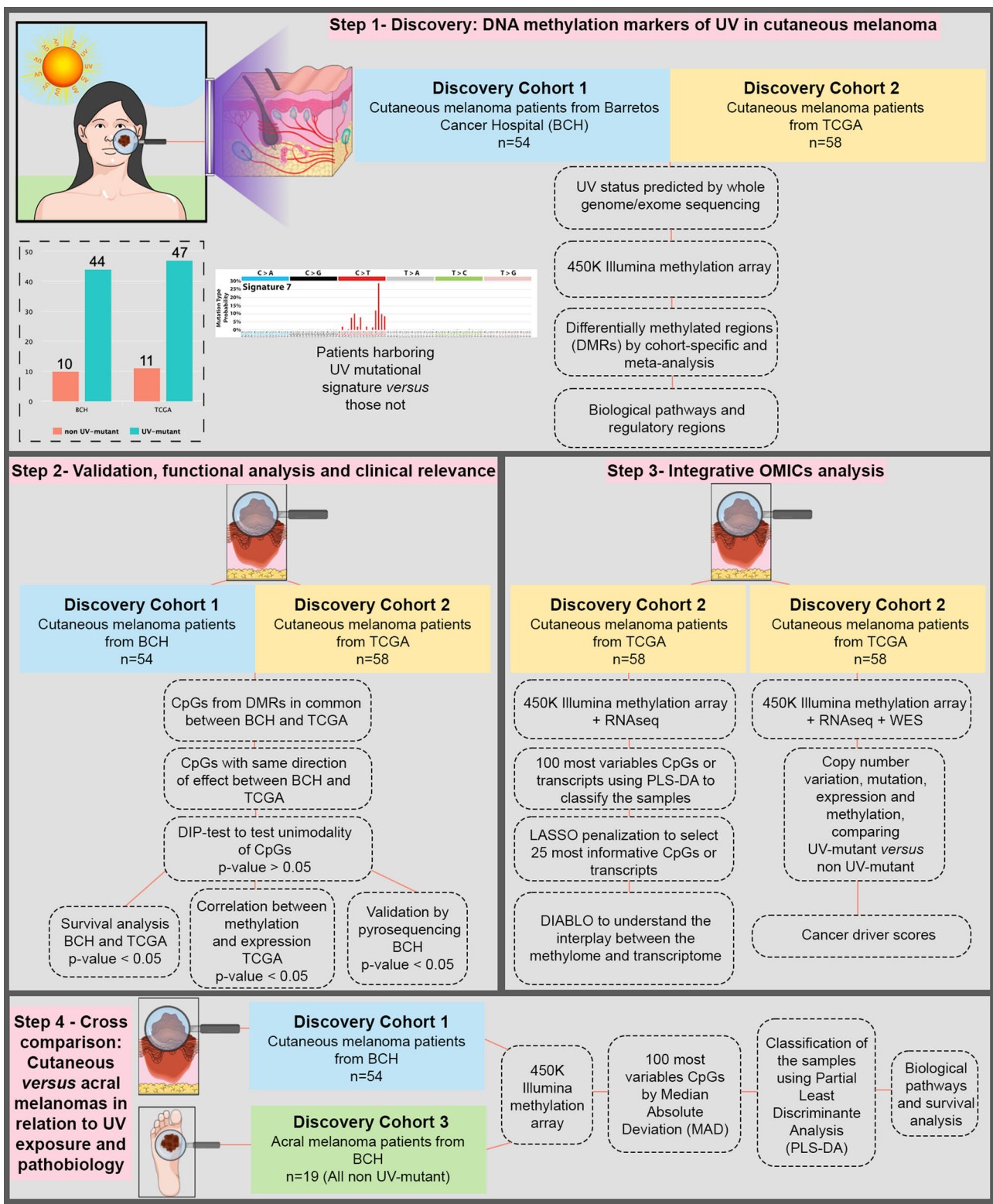

**Fig. 1 Study design, resources and methodology.** A major aim (steps 1–2) is the discovery and validation of genome-wide methylation alterations associated with the UV mutational signature in cutaneous melanoma, based on two independent cohorts. Another major aim (step 3) is assessing the discriminative potential of the DNA methylome *versus* transcriptome *versus* integrated methylome-transcriptome in differentiating between UV-mutant and non UV-mutant cutaneous melanomas. The integrative OMICs approach is expanded to include small nucleotide variants (SNVs) and copy number variants (CNVs) in order to assess cancer driver potential of prioritized differentially methylated genes. This is complemented by step 4, which investigates whether the DNA methylome could capture pathological and/or UV-related differences between major melanoma types predominantly associated with UV exposure (cutaneous melanoma) and those not (acral melanoma).

**Table 1 Clinicopathological characteristics of the BCH-cutaneous melanoma patients, TCGA-cutaneous melanoma patients, and BCH-acral melanoma patients profiled with the DNA methylation array.**

| Characteristics | Cutaneous BCH (Discovery cohort 1) No. of patients (%) | Cutaneous TCGA (Discovery cohort 2) No. of patients (%) | Acral BCH (Discovery cohort 3) No. of patients (%) |
|---|---|---|---|
| Gender | | | |
| Male | 33 (61.1) | 31 (53.4) | 13 (61.9) |
| Female | 21 (38.9) | 27 (46.6) | 8 (38.1) |
| Skin§§§ | | | |
| White | 51 (94.4) | 58 (100.0) | 15 (71.4) |
| Pigmented | 3 (5.6) | 0 (0.0) | 6 (28.6) |
| Tumor type** | | | |
| Primary | 20 (37.0) | 9 (15.5) | 13 (61.9) |
| Superficial spreading | 16 (65.0) | Missing | Not applicable |
| Nodular | 7 (35.0) | Missing | Not applicable |
| Metastatic | 34 (63.0) | 49 (84.5) | 8 (38.1) |
| Superficial spreading | 16 (47.1) | Missing | Not applicable |
| Nodular | 8 (23.5) | Missing | Not applicable |
| Missing | 10 (29.4) | Missing | Not applicable |
| Breslow depth (mm) | | | |
| Up to 1.0 | 5 (9.2) | 5 (8.6) | 2 (9.5) |
| 1.1–2.0 | 7 (13.0) | 14 (24.1) | 2 (9.5) |
| 2.1–4.0 | 9 (16.7) | 9 (15.5) | 3 (14.3) |
| More than 4.0 | 22 (40.7) | 15 (25.9) | 14 (66.7) |
| Missing | 11 (20.4) | 15 (25.9) | Not applicable |
| TNM stage* | | | |
| In situ | 0 (0.0) | 0 (0.0) | 0 (0.0) |
| I | 7 (13.0) | 8 (13.8) | 1 (4.8) |
| II | 16 (29.6) | 18 (31.0) | 8 (38.1) |
| III | 14 (25.9) | 19 (32.8) | 10 (47.6) |
| IV | 15 (27.8) | 2 (3.4) | 2 (9.5) |
| Missing | 2 (3.7) | 11 (18.0) | Not applicable |
| UV signature§§§ | | | |
| No | 10 (18.5) | 11 (19.0) | 17 (81.0) |
| Yes | 44 (81.5) | 47 (81.0) | 4 (19.0) |
| Molecular group**§§§ | | | |
| BRAF | 39 (72.2) | 23 (39.7) | 4 (19.0) |
| RAS | 6 (11.1) | 20 (34.5) | 6 (28.6) |
| NF1 | 2 (3.7) | 6 (10.3) | 1 (4.8) |
| TN | 7 (13.0) | 9 (15.5) | 10 (47.6) |
| Age [Mean (SD)] | 57.3 (17.0) | 57.7 (14.7) | 59.4 (12.5) |

*, §p < 0.05; **, §§p < 0.01; ***, §§§p < 0.001.
*Variables statistically different between BCH-cutaneous and TCGA-cutaneous melanoma patients using two-sided Chi-square test.
§Variables statistically different between BCH-cutaneous and BCH-acral melanoma patients using two-sided Chi-square test.

the *BRAF* mutant group was the most enriched in UV-mutant patients in both BCH and TCGA, though reaching statistical significance only in BCH (Fig. 3a, b). This was in line with other studies[15,16], further reinforcing the reproducibility potential of our data. Interestingly, *BRAF*, *NF1* and *RAS* were not significantly differentially methylated in melanoma tissues in relation to UV exposure (Supplementary Data 7, 8), highlighting that UV exposure produces DNA methylation changes in genes that can be different from critical ones mutationally altered by the same environmental exposure.

Jensen disease analysis of the filtered DMRs showed a significant implication of the differentially methylated genes in skin disorders, such as systemic scleroderma (BCH and TCGA), vitiligo (BCH and TCGA), melanoma (BCH), and skin cancer (BCH), particularly among the top and false discovery rate (FDR)-adjusted ontologies (FDR < 0.05) (Supplementary Data 9, 10). A number of other cancers and diseases were significantly enriched as well (Supplementary Data 9, 10). This was complemented by KEGG pathway analysis, revealing 28 and 30 significant pathways ($p < 0.05$) in BCH and TCGA, respectively. Among them, a large proportion (10 pathways) were identical between BCH and TCGA, 8 and 6 of which remained significant after adjustment for the number of CpGs per gene and FDR, respectively (Fig. 3c, Supplementary Data 11–14). These pathways constituted of differentially methylated genes implicated in immune system regulation: hematopoietic cell lineage, allograft rejection, graft-versus-host disease, intestinal immune network for IgA production, antigen-processing presentation, inflammatory bowel disease, and relatedly, autoimmune diseases, such as type 1 diabetes mellitus, autoimmune thyroid disease, systemic lupus erythematosus and rheumatoid arthritis (Fig. 3c).

The role of DNA methylation alterations in regulating immune system function was investigated in further depth and validated using RNA sequencing data (Methods section), demonstrating that immune cell composition was indeed different between UV-mutant and non UV-mutant cutaneous melanoma patients (Fig. 3d). Specifically, dendritic cells were significantly infiltrated in the non UV-mutant than in UV-mutant cutaneous melanoma (Fig. 3d, $p = 0.03$). Complementary analysis using differentially expressed genes comparing non UV-mutant and UV-mutant cutaneous melanoma patients ($p < 0.05$, Supplementary Data 15) also showed enrichment in immune disorders and skin-related diseases, though none reached FDR significance (Supplementary Data 16, 17).

almost all non UV-mutants analyzed. A similar pattern was observed in the TCGA cohort. Cluster C1 was fully occupied by non UV-mutant samples and was visually distinct, exhibiting again an upper hypermethylation (red) stretch and a lower hypomethylation (blue) stretch, relative to the other clusters. Even if C1 is merged with the adjacent cluster C2, the non UV-mutant patients remain statistically enriched in this combined cluster ($p = 3.00e-10$), which now encompasses almost all non UV-mutants analyzed (Fig. 3a).

Recently, TCGA-cutaneous melanoma patients have been classified into four molecular mutation subgroups: 1—*BRAF*, associated with younger patients and with *BRAF* and *MITF* amplifications; 2—*RAS*, associated with MAPK activation and AKT3 overexpression; 3—*NF1*, associated with older patients and higher mutation burden; 4—The triple negative (TN), which is wild-type for *BRAF*, *RAS*, and *NF-1*, lacks the UV mutational signature and has higher copy number and complex rearrangements[9]. Indeed, we observed that the TN molecular subgroup is significantly enriched in non UV-mutant patients in both BCH and TCGA cohorts (Fig. 3a, b). We also observed that

**DNA methylome markers of UV are prognostic of patient survival.** In addition to the large proportion of overlap in biological pathways between BCH and TCGA described above, there was a significant overlap in DNA methylation alterations at the gene and CpG levels between BCH and TCGA (Fig. 4a). Out of the 458 CpGs from 169 genes significantly overlapping ($p = 2.3e-109$ and $p = 3.71e-29$, respectively) between BCH and TCGA cohorts (Supplementary Data 18, Fig. 4a), 6 CpGs (*HOXC9*, *KCNQ1DN* and *MGMT* genes) were hypermethylated and 30 CpGs (*TAPBP*, *ERICH3*, *FINL2*, *ZNF732*, *SLC6A18*, *MFSD13A*, *SLFN12L*, and *IFNLR1* genes) were hypomethylated in both BCH and TCGA, considering CpGs with absolute effect sizes ≥10% and with no significant enrichment in SNPs (Supplementary Fig. 1b, c).

We complemented the cohort-specific analyses with a DMR meta-analysis across the BCH and TCGA datasets (Fig. 4b). As the results demonstrate, there are 45,915 CpGs significantly differentially methylated across the two datasets between UV-mutant and non UV-mutant cutaneous melanomas (FDR < 0.05), of which a high proportion of CpGs (equal to 24,711 CpGs or equivalent to 53.8%) have the same direction of effect between

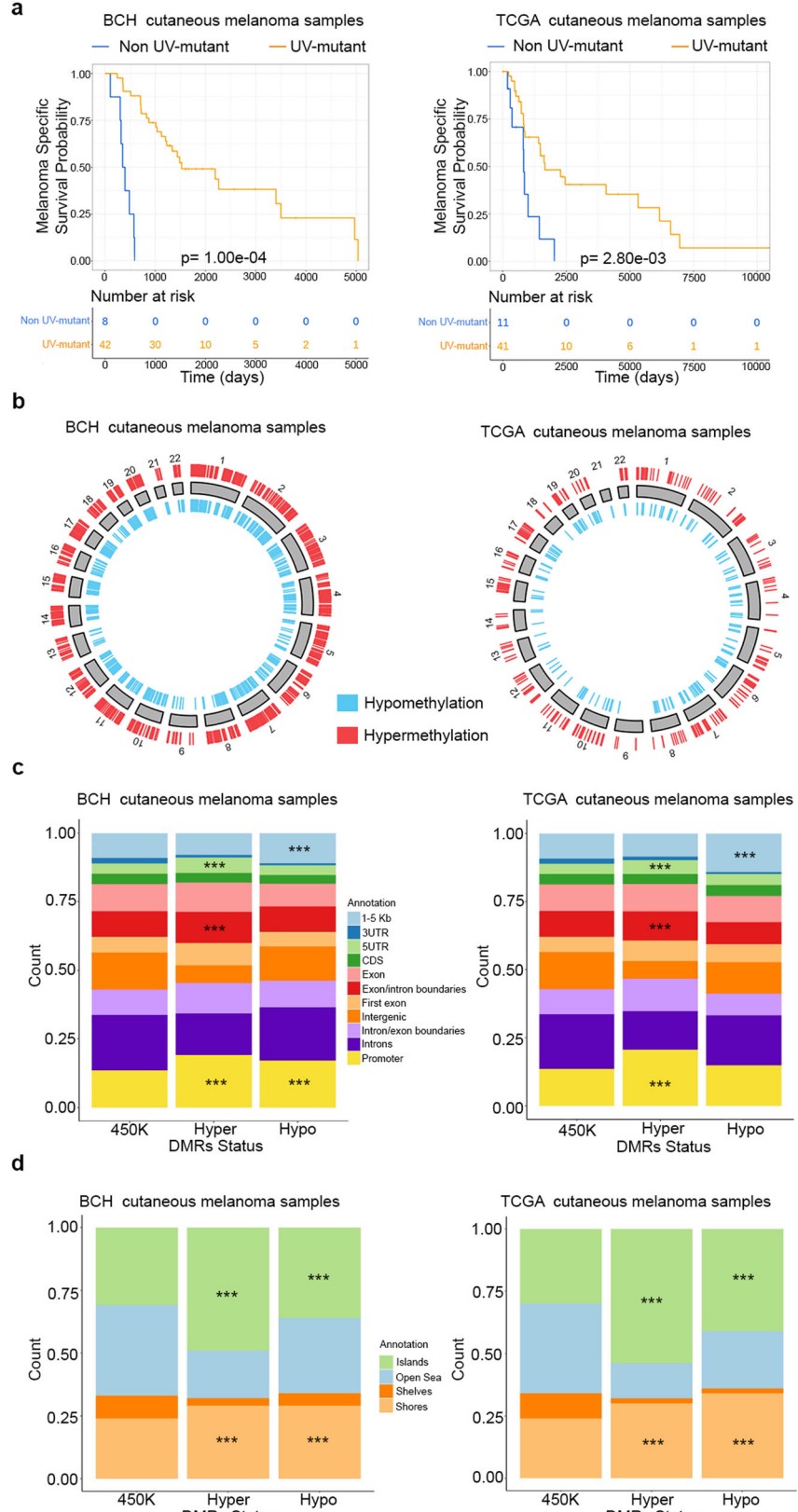

**Fig. 2 Cross genome-methylome analysis of UV mutation signatures in cutaneous melanoma patients from BCH and TCGA cohorts. a** Kaplan–Meier survival curves of melanoma patients by UV signature status in BCH ($n = 50$) and TCGA ($n = 52$). The $P$ values were derived by log-rank test. Also shown are the DMR distributions from the crude model relative to chromosomal location (**b**), genomic regulatory regions (**c**), and CpG density regions (**d**) in both BCH ($n = 54$) and TCGA ($n = 58$). Enrichment analysis of hyper- and hypomethylated DMRs relative to the 450 K reference set in (**c**, **d**) was done using two-sided Chi-square test. ***$p < 0.001$.

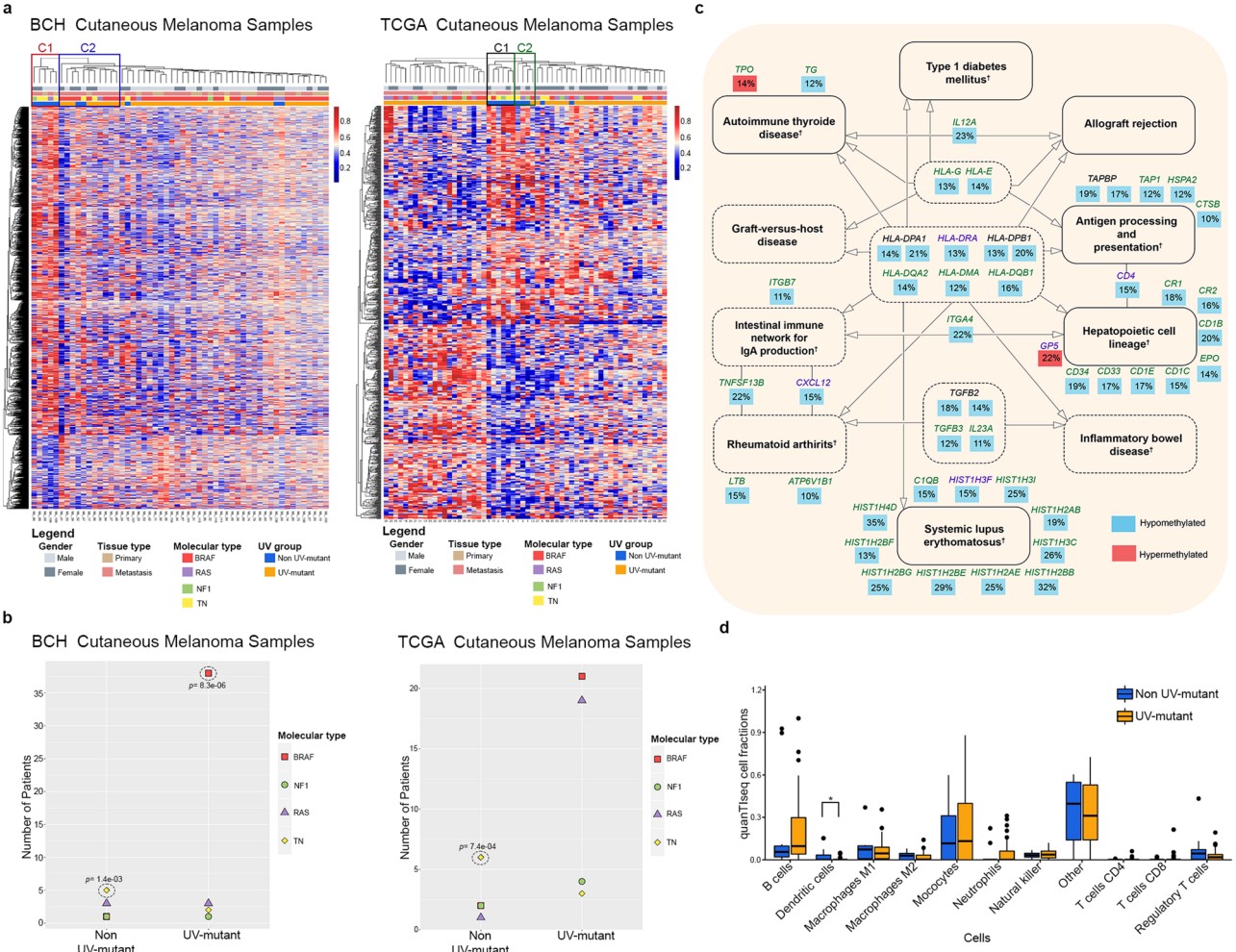

**Fig. 3 The DNA methylome marks UV exposure and associated *BRAF/RAS/NF1* mutations, with effect on immunomodulation. a** Hierarchical clustering of cutaneous melanoma patients in BCH and TCGA based on methylation levels of 4721 and 793 CpGs, respectively, as derived from Supplementary Fig. 1a. Enrichment analysis for non UV-mutant patients in clusters C1–C2 in each cohort was performed using two-sided Chi-square test while delimiting the cluster boundaries by the limits statistically specified by Euclidean distance. **b** Proportions of *BRAF, NF1, RAS*, and TN groups in UV-mutant and non UV-mutant melanomas. *P* values were derived by two-sided Fisher's exact test. **c** Common KEGG pathways between BCH and TCGA of genes differentially methylated between UV-mutant and non UV-mutant cutaneous melanoma patients, as derived from the prioritized CpGs in Supplementary Fig. 1a. Solid lines around the pathways' names indicate those with FDR < 0.05 in BCH and/or TCGA; whereas, dashed lines indicate those with *p* < 0.05. The percentage represents the average effect size across the CpGs of a given gene. Genes written in black are common between BCH and TCGA; whereas, green ones were found only in BCH and purple ones only in TCGA. †Pathways significant after adjustment for the number of CpGs associated with each gene. *P* value was delivered from two-sided Fisher exact test. **d** Immune cell composition inferred from RNA sequencing data comparing UV-mutant (*n* = 47) and non UV-mutant (*n* = 11) cutaneous melanoma patients from TCGA. Box center lines, bound of the box, and whiskers indicate medians, first and third quantiles, and minimum and maximum values within 1.5 × IQR (interquartile range) of the box limits, respectively. Each data point in the box plot represents the samples. *\*p* < 0.05, by two-sided Mann–Whitney *U* Test.

BCH and TCGA (Supplementary Data 19, Fig. 4b). 121 meta-analysis CpGs (FDR < 0.05) overlapped with the 458 CpGs that are common between the BCH and TCGA cohort-specific analyses. As expected, the meta-analysis yields a larger number of significant hits (due to higher statistical power) than the cohort-specific analyses. However, the former is more prone to false positivity especially given some clinicopathological and ethnic dissimilarities (Table 1) and methodological differences between the two cohorts in inferring UV signature status (WGS versus WES, respectively). For this reason, (1) we additionally report the more stringent Bonferroni threshold, which yielded similar results as FDR (Supplementary Data 20, Fig. 4b), and (2) we present the meta-analysis results as a complementary method that reinforces the robustness of the findings across the different cohorts and analysis approaches, while prioritizing the more

conservative cohort-specific analysis which yields signals that are common between BCH and TCGA and which, though less profuse, are less prone to error.

Thus, we further investigated whether the 36 CpGs in common between BCH and TCGA could be used to predict the survival of patients with cutaneous melanoma. Among them, cg06230948-*TAPBP*, cg18930100-*TAPBP*, cg19495013-*FIGNL2*, and cg26835312-*IFNLR1* were significantly associated with survival in BCH after adjustment for multiple testing (FDR < 0.05) (Supplementary Data 21). Among these four CpGs, cg18930100-*TAPBP* was also significantly associated with survival in a lookup analysis in TCGA. Specifically, patients in the low methylation groups at this CpG site had significantly higher melanoma-specific survival in both cohorts (Supplementary Data 21 and Fig. 4c). This was coherent with the hypomethylation at this CpG (Fig. 4d) and increased survival

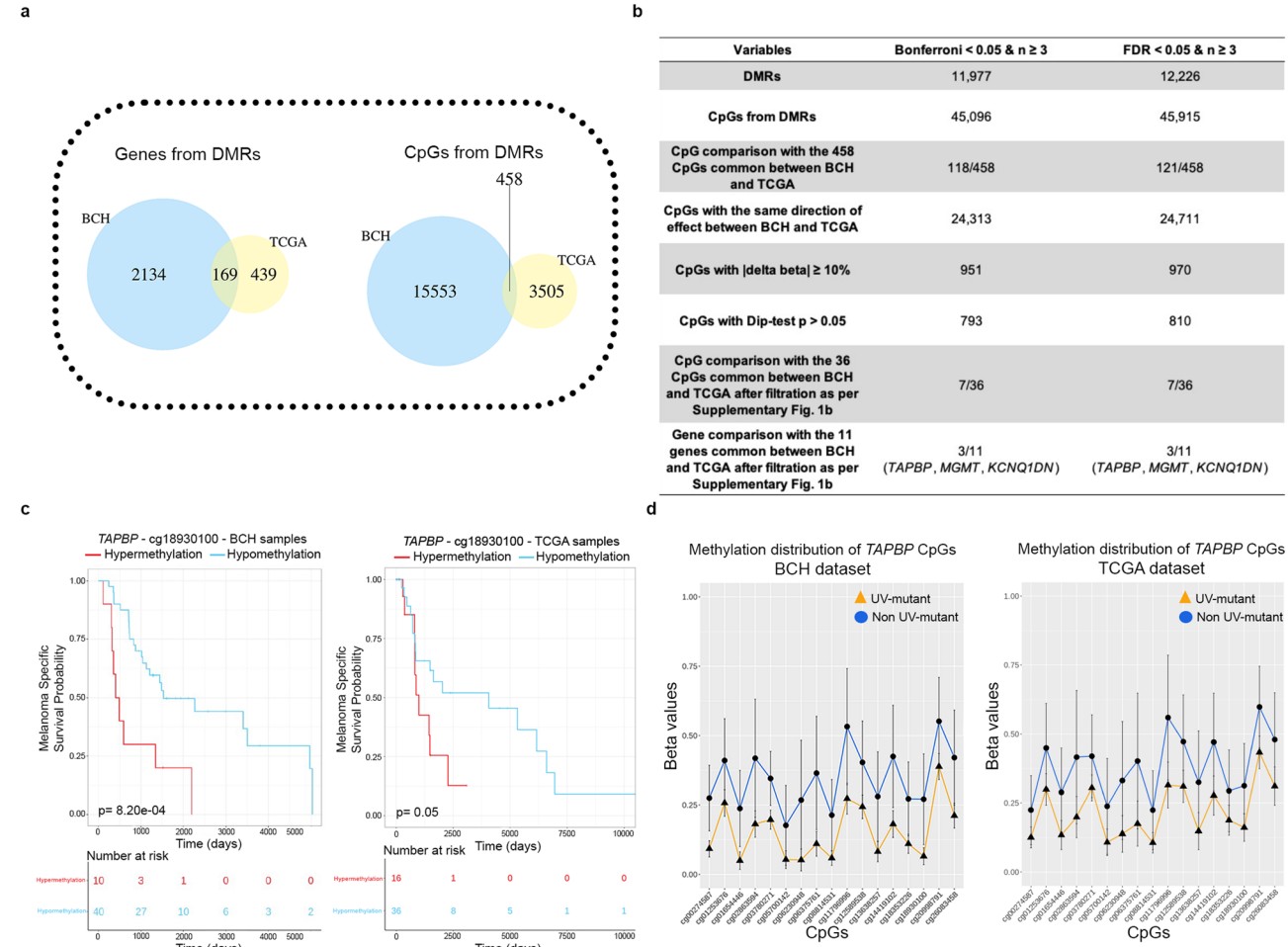

**Fig. 4 UV-related DNA methylome-wide alterations common between BCH and TCGA are prognostic for survival in cutaneous melanoma patients.**
**a** Venn diagrams showing that DMR-derived 169 genes or 458 CpGs are common between BCH and TCGA, based on the crude model. **b** DMR fixed effects inverse variance-weighted meta-analysis of BCH and TCGA, and comparison with the cohort-specific analysis. **c** Kaplan–Meier survival of melanoma patients in relation to methylation levels of cg18930100 (*TAPBP*) measured in the target tumors derived from BCH and TCGA. Patients were categorized into low- and high-methylation groups depending on whether the methylation value of a given CpG is lower or higher, respectively, than the mean methylation across the samples profiled for that CpG. *P* values were derived by two-sided log-rank test. **d** DNA methylation profiles of cg18930100 (*TAPBP*) that is associated with melanoma-specific survival, showing differential methylation between UV-mutant (*n* = 40 and 41 in BCH and TCGA, respectively) and non UV-mutant patients (*n* = 10 and 11 in BCH and TCGA, respectively). Data were expressed as the average values of each group (UV-mutant and non UV-mutant) for each single CpG with error bars indicating the 95% confidence interval (CI).

(Fig. 2a) observed in UV-mutant relative to non UV-mutant patients. Notably, *TAPBP* differential methylation is robustly significant in both the cohort-specific and meta-analyses of the BCH and TCGA cohorts (Fig. 4b and Supplementary Data 7, 8, 19, 20).

**Validation and multi-OMICs functional roles of UV methylome markers.** We next investigated the functional effect of UV-related DNA methylation alterations on gene expression using expression quantitative trait methylation (eQTM) analysis applied to DNA methylome and transcriptome data profiled on the same samples (Fig. 1). We first used a targeted approach focusing on cg18930100-*TAPBP* prioritized in the previous analysis (Fig. 4) and found that its methylation levels were significantly correlated with *TAPBP* RNA expression changes (Fig. 5a). We then performed eQTM analysis on all 458 CpGs that are common between TCGA and BCH (Supplementary Fig. 2a) in order to investigate whether the *TAPBP* gene could be still identified agnostically among the eQTMs. Out of the 458 CpGs, 10 (*TAPBP*: cg01253676, cg01654446, cg06230948, cg06375761, cg02863594, cg18930100,

cg18353226; and *EIF2AK4*: cg20255370, cg16127683, cg01081584) were significantly correlated with expression, among which 7 CpGs were indeed located in the *TAPBP* gene, including its cg18930100 (Fig. 5a). All the significant correlations showed an inverse association between CpG methylation and RNA expression levels of each gene, with *TAPBP* showing hypomethylation while *EIF2AK4* showing hypermethylation in UV-mutant relative to non UV-mutant cutaneous melanoma patients (Fig. 5a). Notably, cg18930100 in *TAPBP* presented hypomethylation associated with both increased *TAPBP* RNA expression (Fig. 5a) and increased patient survival (Fig. 4c) in UV-mutant relative to non UV-mutant cutaneous melanoma patients. *TAPBP* and *EIF2AK4* RNA expression levels did not significantly associate with patient survival (Supplementary Fig. 2b), suggesting that their methylation levels may be stronger prognostic markers than their transcript levels.

Next, we pooled all 36 CpGs prioritized in Supplementary Fig. 1b (being common between BCH and TCGA) with the 10 CpGs prioritized in Supplementary Fig. 2a (being significant eQTMs) and investigated their cancer driver potential derived

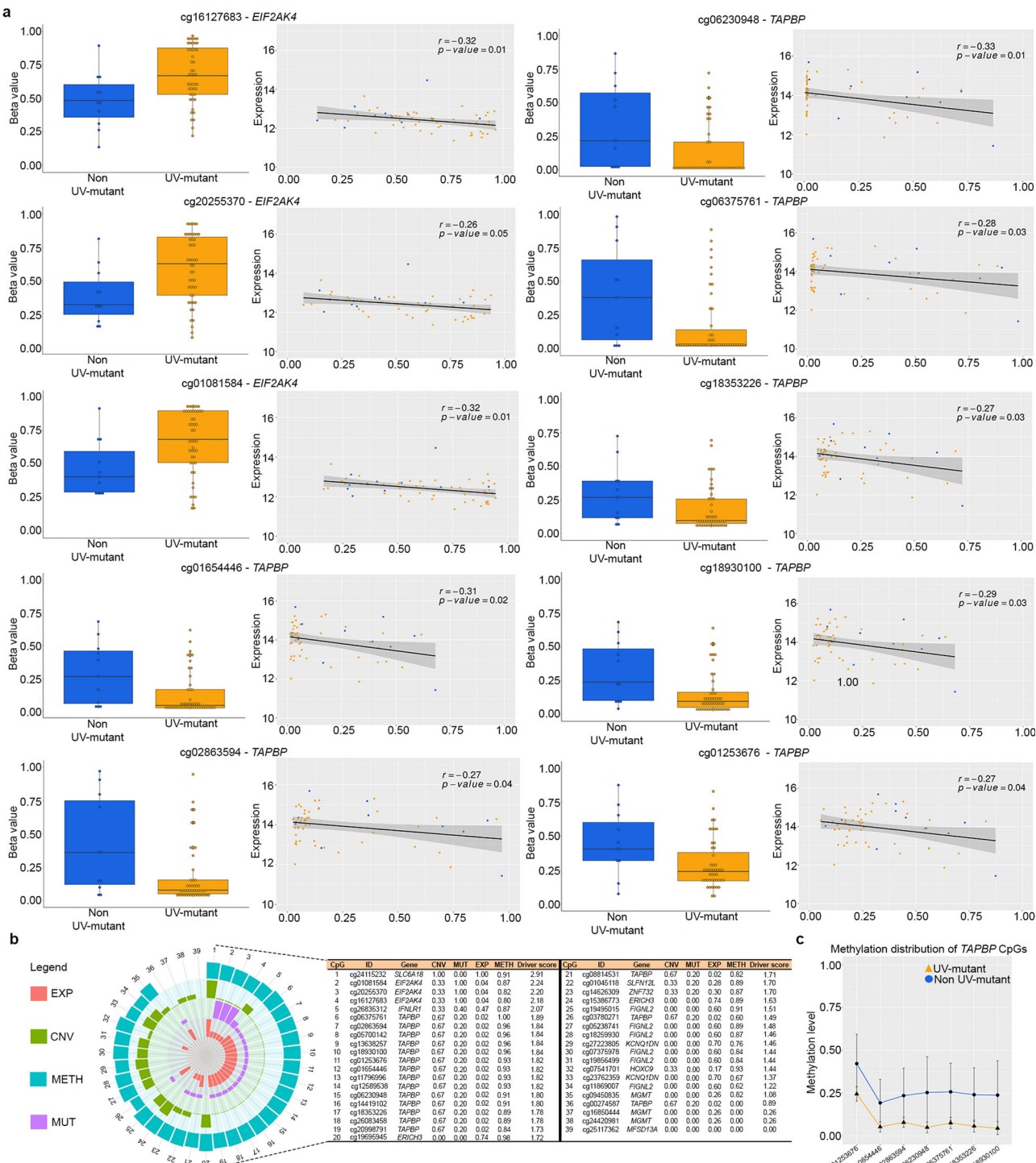

**Fig. 5 Validation, eQTM, and multi-OMICs cancer driver analysis of UV methylome markers. a** Pearson correlation was used to measure linear relationships between DNA methylation (Beta values) and gene expression levels measured in the same samples for the 10 selected CpGs (filtration step described in Supplementary Fig. 1a), using the TCGA dataset (UV-mutant = 11 and non UV-mutant = 47). Box center lines, bound of the box, and whiskers indicate medians, first and third quantiles, and minimum and maximum values within 1.5 × IQR (interquartile range) of the box limits, respectively. Each data point in the box plot represents the samples. The correlation $r$ and $P$ values were calculated by the two-sided correlation test and are shown for each CpG. **b** Multi-OMICs data integration from TCGA, encompassing copy number variation (CNV), expression (EXP), methylation (METH) and mutation (MUT), was performed in order to decipher the driver potential of the 12 prioritized genes (see Results) in cutaneous melanoma development following UV exposure. For each gene, scores of CNV, MUT, EXP, METH and multi-OMICs driver are indicated in the table and plotted in the associated circular diagram. **c** Validation of array-based DNA methylation by bisulfite pyrosequencing of the *TAPBP* gene in BCH samples (UV-mutant = 7 and non UV-mutant = 40). Data were expressed as the average values of each group (UV-mutant and non UV-mutant) for each single CpG with error bars indicating the 95% confidence interval.

from our recent multi-OMICs driver score[17]. This was performed using data on copy number variation, point mutations, RNA expression and DNA methylation profiled in cutaneous melanoma patients. We found that the top half of the CpGs with the highest cancer driver potential were largely predominated by CpGs of the *TAPBP* gene (Fig. 5b) and that this gene ranked among the top 4 driver genes when methylation levels were averaged across CpGs of a given gene (Supplementary Fig. 2c).

As a positive control, we used a list of genes known to play driver roles in cutaneous melanoma based on the Consensus-Driver score method (i.e., with Consensus-Driver >1.5)[18], which preferentially selects cancer driver genes that are frequently mutated in tumor tissues. We calculated the multi-OMICs driver scores for those genes, derived by measuring the extent of their OMICs alterations in UV-mutant relative to non UV-mutant melanomas (Supplementary Fig. 2d), as was done for the experimental gene set (Fig. 5b). We found that the multi-OMICs driver scores of the latter, including *TAPBP*, were predominantly in the same range as that of the positive control genes (1.24–2.50) (Supplementary Fig. 2d), reinforcing the cancer driver potential of the experimental gene set relative to known driver genes in melanoma.

Because of the biological and clinical relevance of *TAPBP* methylation, which was correlated with melanoma-specific survival and RNA expression and concurred with other genome-wide deregulations that led to its high multi-OMICs driver potential, we performed technical validation of *TAPBP* methylation using bisulfite pyrosequencing in the BCH cohort (Supplementary Data 22). Methylation by pyrosequencing validated that obtained with the methylome-wide array, confirming the observed *TAPBP* hypomethylation (including similar effect sizes and baseline methylation levels) in UV-mutant relative to non UV-mutant cutaneous melanoma (Fig. 5c).

**Cutaneous and acral melanoma cross-OMICs: UV *versus* pathobiology**. In addition to the genome- and methylome-wide analysis of UV exposure status in cutaneous melanoma, we next investigated whether the transcriptome landscape, taken alone or integrated with the methylome map, can better distinguish UV-mutant from non UV-mutant melanomas (Fig. 1). Based on PLS-DA modeling (Methods section), we observed that the DNA methylome alone predicts the two groups of patients (Fig. 6a—left panel) better than the transcriptome alone (Fig. 6a—right panel). The discriminative potential of the DNA methylome between UV-mutant and non UV-mutant cutaneous melanomas was sufficiently powerful, with slight or no improvement observed by the integrated methylome-transcriptome map (Fig. 6b and Supplementary Data 23) using LASSO coupled to DIABLO (Methods section). This complemented our earlier results showing that DNA methylation levels altered in UV-mutant melanomas are more prognostic to patient survival than the transcript levels of the corresponding genes (Fig. 4 and Supplementary Fig. 2b).

We further investigated whether the DNA methylome could also underlie differences between pathologically different melanomas, with interaction by UV mutation status, namely between melanoma types predominantly associated with UV exposure (cutaneous melanoma) and those not UV-associated (acral melanoma). Table 1 shows the clinical annotations of the acral samples collected at BCH. In contrast to cutaneous melanomas, out of 21 acral, only a few (19.0% compared to 81.5% in BCH-cutaneous melanoma, $p = 4.24e{-}07$) had the UV mutation signature as expected. The majority (47.6%) of the acral melanoma patients did not exhibit mutations in *BRAF*, *NRAS* or *NF1*, and a substantial portion (28.6% compared to 5.6% in

BCH-cutaneous melanoma, $p = 2.28e{-}05$) presented a pigmented skin phenotype.

Based on PLS-DA modeling in BCH, we observed that the DNA methylome of the non UV-mutant cutaneous melanomas resembles more that of the pathologically different acral melanomas than the pathologically related UV-mutant cutaneous melanomas (Fig. 6c). This was in line with the survival analysis showing that the non UV-mutant cutaneous melanoma patients presented worse prognosis, more closely resembling that of acral melanoma patients (known to have poorer prognosis) rather than that of UV-mutant cutaneous melanoma patients (Fig. 6d, $p < 1.00e{-}04$).

The impact of tumor pathology on DNA methylome alterations was still observable, however, as evident by 1784 DMRs distinguishing non UV-mutant cutaneous and non UV-mutant acral melanomas (Supplementary Data 24). Jensen disease ontology and KEGG pathway analyses of the filtered DMRs (as described in Supplementary Fig. 2e and Supplementary Data 25) showed a significant implication ($p < 0.05$) of the differentially methylated genes in skin disorders (Supplementary Data 26) and immunological pathways (Fig. 6e, Supplementary Data 27, 28), respectively, but we interpret these results with caution as none of them remained significant after correcting for multiple testing (FDR > 0.05) (Supplementary Data 26–28).

## Discussion

Melanoma is a type of skin cancer, which represents one of the most complex and heterogeneous cancers compared to other cancer types[19]. Although the positive association between UV exposure with melanoma development is well known, the underlying epigenetic mechanisms are largely unexplored in human melanoma tissues, as outlined by our systematic literature search (Supplementary Data 1). With the advent of new powerful technologies, such as WGS/WES, UV exposure status can now be inferred and analysed in human tissues. The present study investigated DNA methylome-wide alterations associated with UV mutation status in two cohorts of human cutaneous melanomas, with in-depth analysis of the functional and clinical implications of those alterations, including effects on regulatory regions, biological pathways, gene transcription, cancer driver potential, tumor classification and patient survival. This was complemented by testing whether the DNA methylome could also underlie differences between pathologically and molecularly different subtypes of melanomas with interaction by UV mutation status, namely between BRAF, RAS, NF1 and TN molecular groups and between melanoma types predominantly associated with UV exposure (cutaneous melanoma) and those not UV-associated (acral melanoma). To date, there is only one study that described the methylome landscape of acral melanomas[20], and our work additionally highlights genes and biological pathways that are epigenetically deregulated in this uncommon melanoma type in comparison with cutaneous melanoma analysed in the same cohort encompassing patients of European and Latin-American descents.

The only available melanoma dataset with methylome and genome data for replication of our BCH findings was from TCGA. The number of detectable signals was higher in the BCH relative to the TCGA cohort, and this is probably not due to statistical power differences as both datasets had similar sample sizes. This could be rather due in the BCH dataset to (1) the better quality of samples and/or their processing using our in-house optimized automated workflow to generate DNA methylome data, coupled to a priori designed sample distribution on the array that minimizes confounding with batch effects based on statistical semi-randomization, and (2) more accurate technology,

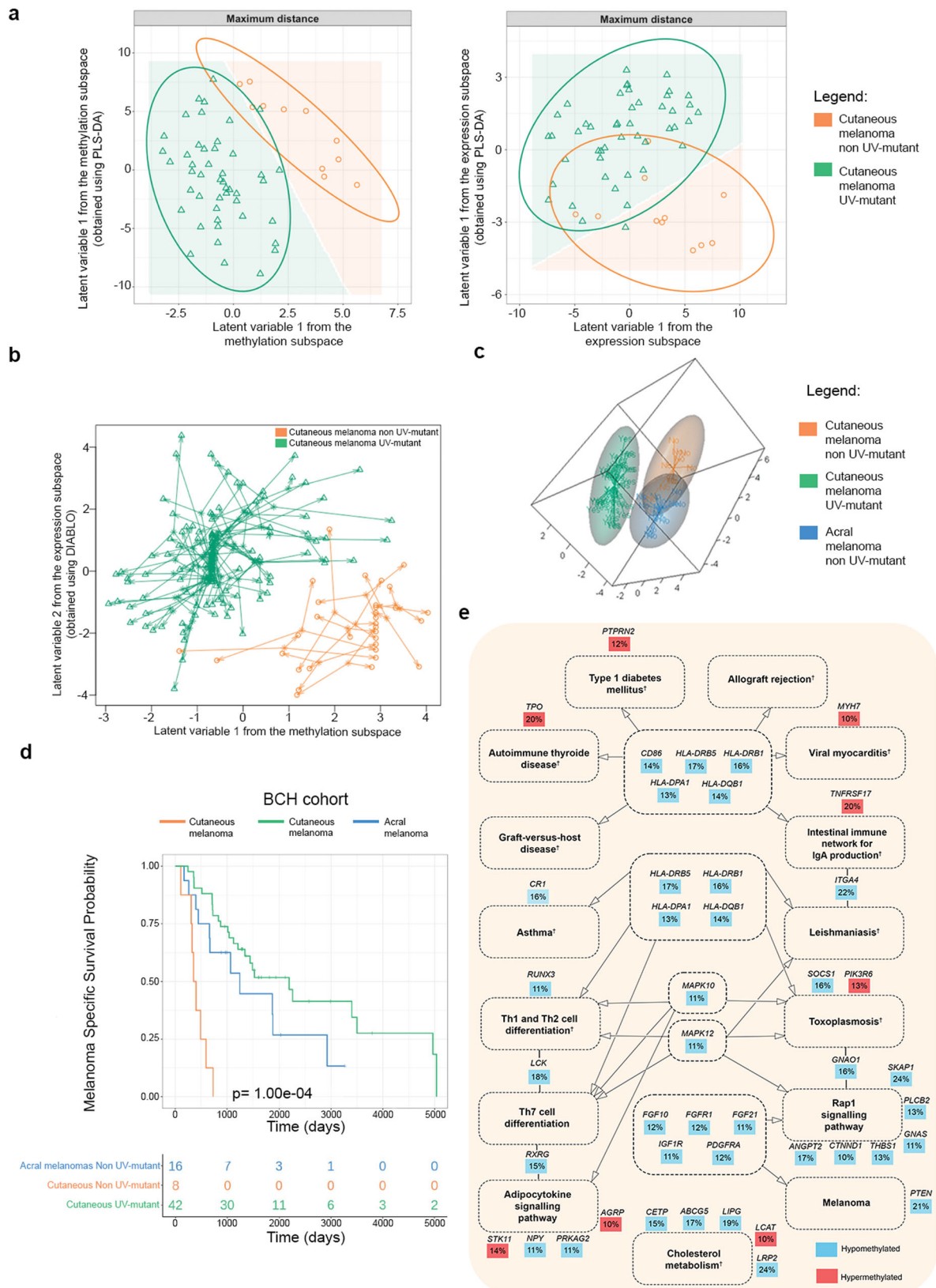

using WGS rather than WES, to assess UV exposure. Even though there were some clinicopathological and ethnic dissimilarities (Table 1) and methodological differences between the two cohorts in predicting UV signature status (WGS *versus* WES, respectively), we observed consistent findings in both at the CpG, gene and biological pathway levels. Moreover, UV-related DNA methylation alterations showed similar distributions between BCH and TCGA in hypo- and hypermethylated regions as well as similar enrichments in regulatory and CpG density regions, in skin disorders and in immunological pathways. Among the CpGs and genes differentially methylated in both BCH and TCGA, methylation levels of *TAPBP* (encompassing several differentially

**Fig. 6 Comparative maps of the DNA methylome and transcriptome of cutaneous and acral melanomas in relation to UV mutational signatures, with associated patient survival and biological pathways. a** PLS-DA modeling based on DNA methylome (left panel) or transcriptome (right panel) data derived from UV-mutant and non UV-mutant cutaneous melanoma from TCGA. **b** Diablo integrative analysis method with LASSO penalization were applied on methylome and transcriptome data from TCGA to select the most informative CpGs and transcripts that could improve the classification of the UV-mutant and non UV-mutant cutaneous melanomas. **c** Methylome matrices of acral (excluding the few UV-mutants), UV-mutant and non UV-mutant cutaneous melanomas based on the 100 most variables CpGs selected using median absolute deviation (MAD) and analysed with Partial Least Squares Discriminant Analysis (PLS-DA) in the BCH samples. **d** Melanoma-specific survival comparing the three groups of melanoma patients: cutaneous UV-mutant ($n = 42$), cutaneous non UV-mutant ($n = 8$) and acral non UV-mutant ($n = 16$). $P$ value was delivered from log-rank test. **e** KEGG pathways of genes differentially methylated between acral *versus* cutaneous non UV-mutant melanomas in BCH, as derived from the prioritized CpGs in Supplementary Fig. 2a. Dashed lines around the pathways' names indicate those with $p < 0.05$; none were FDR-significant. The percentage represents the average effect size across the CpGs of a given gene. †Pathways significant after adjustment for the number of CpGs associated with each gene.

---

methylated CpGs) were significantly associated with RNA expression of this gene, concurred with other genome-wide deregulations to yield a high multi-OMICs driver potential and were significantly correlated with melanoma-specific survival. The array-based methylation results of *TAPBP* were independently validated by bisulfite pyrosequencing, further reinforcing the robustness of the findings and providing promising opportunities for clinical application via pyrosequencing as a cost-effective technique.

*TAPBP* is a member of the immunoglobulin superfamily, which mediates the interaction between newly assembled major histocompatibility complex class 1 (MHC-I) and the transporter associated with antigen processing[21]. Downregulation of TAPBP (tapasin) protein expression has been observed in multiple cancers as an immune escape mechanism of human tumors, which is restored after cytokine administration, indicating that deficient TAPBP expression might be due to dysregulation than to structural alterations[22]. Our findings show that *TAPBP* transcription is significantly inversely associated with its DNA methylation levels, and the latter are altered in relation to UV exposure rather than to melanoma pathological identity since this gene was not found to be differentially methylated in non UV-mutant cutaneous *versus* acral melanomas (Supplementary Data 25).

MHC-I complex expression on tumor cells has been described as an excellent surrogate marker of the overall tumor immunogenicity level as well as a predictor of response to immune checkpoint blockade therapy[23]. Moreover, MHC-I down-regulation was identified as a common mechanism of resistance to PD-I inhibitor in melanoma clinical samples[24]. Restoring TAPBP expression can enhance MHC-I (HLA-B and -C) expression, as demonstrated in vitro, highlighting the possibility that patients with defects in MHC-I antigen-processing machinery may benefit from combining immunotherapeutic strategies with demethylating agents (such as those that could restore TAPBP expression)[25]. In complement with *TAPBP*, several HLA genes were also differentially methylated in UV-mutant *versus* non UV-mutant cutaneous melanoma in both BCH and TCGA, and these genes were centrally involved in the multiple immunological pathways identified (Fig. 3c). Taken together, *TAPBP* and MHC-I machinery genes, dysregulated by DNA methylation mechanisms as observed in our study, represent promising targets for epigenetic therapy and for predicting clinical response to immunotherapy.

*TAPBP* methylation significantly predicted patient prognosis in both BCH and TCGA. Even though the sample size of expression data was the same as that of methylation data, the association between TAPBP expression and survival was not significant. This suggests that the difference is not merely due to statistical power but could indeed reflect biological basis. DNA methylation does not act solely through affecting gene transcription but is known to also associate with chromosomal instability, the induction of splice variants, alterations in enhancer regions, changes in

microRNA binding regions and expression control regions, and mutations[26–28]. Hence, DNA methylation may function as a prognostic marker per se or through these various non expression-related mechanisms. Our observation is in line with a multitude of studies highlighting the high sensitivity of the epigenome to exposure and risk factors[12,29].

The relation between *TAPBP* methylation and survival may not be necessarily causal. Our results, however, pinpoint to an increased likelihood of causality because (1) they were reproduced in two independent populations, including different ethnicities, which offer a natural means of effect randomization (hence, minimizing the influence of confounders), (2) they showed a dose-response (*TAPBP* hypomethylation was associated with increased survival relative to hypermethylation), and (3) they yielded a cancer driver potential for *TAPBP* that was comparable to that of known cancer driver genes. Still, more datasets will be needed to better reinforce the causality of the associations, for example, by using in larger sample sizes germline data as proxies for *TAPBP* methylation through Mendelian Randomization.

In addition to the findings focused on *TAPBP*, we reported that the UV mutational signature is associated with a high load (thousands) of epigenetic alterations affecting the methylome landscape of cutaneous melanoma. This highlights that, even though the UV-mutant and non UV-mutant cutaneous melanomas are supposed to share the same pathological/cellular origin, they may need to be classified separately, at least based on their underlying epigenomic landscape, which has the potential to capture markers of both exposures and cell identity. Moreover, the non UV-mutant cutaneous melanoma, by resembling in its epigenome the acral melanoma, may have a poorer prognosis and require a different therapeutic approach than the UV-exposed cutaneous melanoma. This is in line with our data showing that patient survival is worse for the non UV-mutant cutaneous and acral melanomas relative to the UV-mutant cutaneous melanomas. Our findings also corroborate those of another study showing that cutaneous melanoma patients harboring the UV mutation signature had higher disease-free and overall survival[11]. The consistency between findings is notable especially that our dataset included a different ethnic group and a more accurate methodology (WGS rather than WES) to infer the UV signature.

A recent study investigated the genetic changes related to UV in clinical melanoma samples and found 10 genes commonly mutated in UV-mutant relative to non UV-mutant cutaneous melanoma[11]. Among them, *PKHD1L1*, *LRP1B*, *ADGRV1* and *DNAH10* were hypomethylated in UV-mutant compared with non UV-mutant cutaneous melanoma patients in BCH (Supplementary Fig. 3a). Moreover, methylation of 6 CpGs of *LRP1B* were significantly associated with melanoma-specific survival (FDR < 0.05) (Supplementary Data 29); the most significant of those CpGs was cg02322989, the hypomethylation of which was associated with higher melanoma-specific survival (Supplementary Fig. 3b). The TCGA cohort did not corroborate the DMRs of these genes.

Consensus driver[30] and secondary driver genes have been recently described in cutaneous melanoma[31], among which several were differentially methylated in UV-mutant *versus* non UV-mutant cutaneous melanomas patients in our analysis (*COL5A1, DACH1, MECOM, PTEN, TP53, BRD9, BCL7, SPRED1, SIGLEC12,* and *SIGLEC10*) (Supplementary Data 2–6). These driver genes were derived mostly based on genomic data. We identified driver genes in melanoma and validated others already described, based on our multi-OMICs driver score encompassing genomics, transcriptomics and DNA methylome data. Our results suggest that genes differentially methylated in response to UV may play driver mechanisms in melanoma development.

In this work, we applied multiple powerful technologies, encompassing WGS, WES, RNA sequencing and DNA methylome-wide profiling, coupled to state-of-the-art bioinformatics tools onto a unique series of cutaneous and acral melanoma samples. Specifically, we leveraged publicly available data and complemented that with the generation of new datasets, with larger sample sizes, higher genomic coverage, more detailed phenotypic assessment, high-quality frozen tissue samples, and the inclusion of melanomas other than cutaneous and of ethnicities besides European-descent. In fact, less than 5% of genetic studies worldwide include participants with multiple ethnicity[32], specifically in acral melanoma research[33], and our work helps address this timely advocated need[32,33] by contributing to genomics and epigenomics data from populations of non-European descent. By investigating epigenetic markers of UV exposure in human melanoma tissues from two distinct populations and overlaying the DNA methylome landscape onto the transcriptome and genome maps of UV-mutant cutaneous relative to non UV-mutant cutaneous and acral melanomas, this work contributes to (1) uncovering potentially powerful exposure and cancer epigenetic biomarkers that can be exploited in risk stratification; (2) enhancing tumor classification within and across melanoma types; (3) revealing molecular drivers in melanomagenesis that could be at the origins of this cancer, hence, suitable for targeted therapy; and (4) diminishing population disparities and knowledge inequalities in melanoma pathobiology. The translational impact of the work covers common and less frequent melanomas and offers a roadmap guiding similar gene-environment investigations of other melanoma types.

## Methods

### Patient eligibility, biospecimen, and clinical data
The study was conducted according to the Brazilian national and institutional ethical policies, and it was previously approved by the Barretos Cancer Hospital Ethics Committee (716/2013). No compensation was provided to the participants in this study and informed consent was obtained by all participants included in BCH cohort. Patients were recruited at BCH in the context of the ICGC-Brazil project, which encompassed 100 melanoma patients prior to any systemic treatment and from whom paired tumor/blood tissues were profiled by WGS[34] and tumor tissues by 450 K DNA methylation array. We selected two subsets of patients from ICGC-Brazil (BCH cohort): first, Discovery cohort 1, encompassing 54 cutaneous melanomas patients harboring or not the UV mutation signature; second, Discovery cohort 3, constituting of 17 acral melanomas that are non UV-mutant (Fig. 1), after having excluded the 4 acral samples that were UV-mutant. All BCH samples were fresh frozen. Clinicopathological data were collected under ICGC guidelines. During the admission process at BCH, all patients self-report their skin type and ethnicity, and this information was extracted from medical records given the retrospective nature of the study. In addition, several studies were conducted on this patient population to determine their genetic-based ethnicity and correlate their ethnicity with clinical characteristics. These studies observed considerable admixture in the genetic composition[35–38].

The second cohort comprised 58 cutaneous melanoma samples from TCGA-SKCM for which information about UV signature was available (Fig. 1) based on WES. We excluded formalin fixed paraffin embedded samples and selected only fresh frozen samples for best quality of data and to eliminate sample processing bias in our comparisons with BCH samples. Clinicopathological data were downloaded from the TCGA-SKCM published study[9].

### DNA isolation
DNA from fresh frozen BCH-cohort samples were isolated using the DNA Mini Qiasymphony kit (Qiagen catalog no 937236) following BCH Biobank procedures and the manufacturer's instructions[39]. Briefly, ~25 mg tissue in

180 μL ATL Buffer was homogenized (Precellys, Bertin-instruments) at 6500 $1 \times 10/10$ s for three times. After samples were centrifuged for 1 min at $2867 \times g$, supernatants were transferred for another tube, 25 μL of Proteinase K were added per sample, and samples were incubated at 56 °C, $134 \times g$ for 3 h. Then 4 μL RNase were added per sample and DNA was isolated using QIAsymphony (Qiagen catalog no 9001297).

### Whole genome and exome sequencing and inference of UV mutational signatures
The WGS library construction and sequencing of BCH samples were performed at Mendelics (São Paulo, SP, Brazil). A total amount of one μg of each matched normal and tumor DNA was submitted to sonication fragmentation and further library preparation by Illumina TruSeq DNA PCR-Free Library Preparation kit (Illumina catalog no 20015963) using the 350 bp protocol. Libraries were quantified by Qubit Fluorometer (Thermofisher catalog no Q33238) and qualified by 2100 Bioanalyzer (Agilent catalog no G2939BA). The sequencing was carried out using Illumina HiSeq 2500 by paired-end strategy at a minimum of 30X coverage. WES data of TCGA samples was available from GDC Legacy Portal[9]. Molecular subgroups were defined by investigating somatic single-nucleotide mutations in *BRAF* hotspot, *RAS* hotspot and *NF1* throughout Mutect[40] algorithm and further annotated by Annovar[41]. The TN molecular subgroup denoted melanoma patients who did not harbor mutations in any of the three genes[34].

The UV mutational signature identification was performed using the SomaticSignatures Bioconductor package[42]. We used the Non-negative Matrix Factorization (NMF) algorithm[43] to determine the consensus signatures among the 71 patients. At the moment of the analyses, we used the 21 signatures[10] that were available and identified a consensus signature with more than 0.8 cosine similarity. For both BCH and TCGA cohorts, we classified samples as harboring an UV mutation signature (Cosmic Signature 7) based on the recommended criteria in which C > T transitions at dipyrimidine sites accounted for more than 60% or CC > TT mutations more than 5% of the total mutation burden[9].

### Bisulfite conversion
The isolated DNA (500 ng) from BCH-cohort was bisulfite-modified using the EZ DNA Methylation Kit (Zymo Research catalog no #ZD5004) following the manufacturer's instructions for Illumina Infinium 450 K beadchip assay. Modified DNA was stored at −20 °C when short intervals were required between bisulfite conversion and further processing, and at −80 °C for long-term storage.

### 450K DNA methylome-wide array and analysis
The 450K data of BCH were generated in-house, and those of TCGA were downloaded from the GDC Legacy Portal[9]. For BCH, bisulfite converted DNA samples were profiled using 450K (Illumina catalog no WG-314-1003) and a well-established workflow optimized at IARC for high-throughput analyses through an automated robotic system (Freedom EVO 150 by Teca) that can process the chips with minimal human error. Chips are scanned using Illumina iScan to produce two-color raw data files (.idat format). Sample allocation to the arrays was based on a semi-randomization design that ensures minimum confounding by technical variation and minimizes the masking of biological covariates of interest by batch effects.

For the bioinformatics pre-processing, IDAT files from both cohorts were imported and processed using R software. Quality-control graphs and bimodal distributions for each dataset are shown in Supplementary Figs. 4, 5. We excluded cross-reactive probes and XY chromosomes, leaving a total of 459,761, 459,770, and 459,768 probes for the analysis in BCH-Cutaneous, BCH-Acral and TCGA cohort, respectively. The data were further normalized using the FunNorm function of the Bioconductor Minfi package[44] (Supplementary Fig. 4), which was shown to perform equally good or outperform existing normalization methods[45]. Inferred beta values were used to predict sex as a quality-control step using the Minfi function getSex. All samples were correctly predicted. The DNA methylation level β-values were logit transformed to M-values to map the range (0, 1) to (−inf, +inf) as it is more suitable for running regressions. Surrogate variable analysis (SVA) was performed on the methylome data to correct for potential batch effects, to adjust for differences in cell type composition as a reference-free method[46], and to adjust for latent variables, a choice validated by the findings of our benchmarking[47]. SVA also increases statistical power by removing (unwanted) variability through aggregating information at the data level and constraining the data's variability to the phenotype of interest[48].

For the statistical analysis, we used robust linear regression (robust to outliers) to test four statistical models in the discovery cohort 1 (BCH), including one crude and three adjusted models, comparing the DNA methylome of UV-mutant *versus* non UV-mutant cutaneous melanoma patients (Supplementary Figs. 5, 6): 1—Crude Model (Supplementary Data 2); 2—Adjusted Model 1 adjusted for sex (Supplementary Data 3); 3—Adjusted Model 2, adjusted for sex + age at diagnosis (Supplementary Data 4); and 4—Adjusted Model 3, adjusted for sex + age at diagnosis + tumor type (primary or metastasis) (Supplementary Data 5). Supplementary Fig. 5 shows quantile-quantile (Q–Q) plots of −log₁₀P values, which deviate from their expected values under the null hypothesis across all models. Although the adjusted models yielded a larger number of significant findings relative to the crude model (Supplementary Fig. 6), we preferred to take a conservative approach and focus our analysis on the crude model, especially that it

showed the least genomic inflation and risk of false positives, with a lambda of 1.20 (i.e., approaching to 1.0 being the no inflation limit) (Supplementary Fig. 5). Moreover, the predominant proportion of significant findings in the crude model was actually common with any of the adjusted models (Supplementary Fig. 6b). We also compared for each model, two approaches of methylome-wide analysis: the Differentially Methylated CpG Probes (DMPs), analysing individual CpGs using the Bioconductor limma package[49], and the DMRs, analysing regions of genomically proximal CpGs using the Bioconductor DMRcate[50] package with the default proximity-based criteria (±1000 base pairs). At least 90% of DMP-based genes overlapped with those derived from DMRs across all models (Supplementary Fig. 6a). For this reason and because DMR analysis represents a dimension reduction approach with higher statistical power than DMP analysis, we focused downstream analyses in BCH and TCGA data onto the DMR approach applied to the crude model. This pipeline was equally applied to the DNA methylome comparison between non UV-mutant acral and non UV-mutant cutaneous melanomas. Statistically significant DMPs and DMRs were defined as those with FDR-adjusted $P$ value < 0.05.

We complemented the cohort-specific analysis by a meta-analysis across the BCH and TCGA cohorts comparing UV-mutant *versus* non UV-mutant cutaneous melanoma patients. We used the Metal tool[51] and the Dmrff package in R to perform DMR fixed effects inverse variance-weighted meta-analysis[52], using the crude model as prioritized in the cohort-specific analysis. The meta-analysis lambda value was 1.16, showing low inflation. Statistically significant DMRs were defined as those with FDR-adjusted $P$ value < 0.05. Due to the larger number of hits expected with the increased statistical power afforded by the meta-analyses, we also reported the more stringent Bonferroni-adjusted $p$ values, especially considering the higher likelihood of false positivity due to clinicopathological, ethnic and methodological differences between meta-analysed BCH and TCGA.

In addition to generating DMPs and DMRs, methylation data from BCH and TCGA samples were further investigated using Partial Least Squares Discriminant Analysis (PLS-DA)[53]. This approach performs classification of samples using partial least squares regression of the categorical outcome Y (cancer subtype) on the predictor variables (DNA methylation). PLS-DA is a clustering technique that allows the quantification of the discrimination relevance of a given variable (CpG) and to predict the phenotype of new samples (independent of DMPs or DMRs). This method is especially suited to deal with a much larger number of variables than samples, as in next-generation microarray and sequencing data, and we aided this method further by a filtering step using median absolute deviation (MAD)[54]. We selected the 100 most variable CpGs and applied PLS-DA on the methylation matrix on this subset of sites to assess the discriminative potential of the DNA methylome between UV-mutant and non UV-mutant cutaneous melanomas (as well as acral melanomas in the case of BCH samples).

**Pyrosequencing methylation analysis**. For the quantitative measurement of DNA methylation levels in individual CpG sites of the *TAPBP* (7 CpGs) gene (Supplementary Data 22), we pyrosequenced the bisulfite converted DNA using the PyroMark Kit (Qiagen catalog no 978703) as per the manufacturer's instruction. Briefly, DNA was immobilized onto streptavidin-coated beads in binding buffer for 10 min. The biotin-labeled PCR template was isolated and denatured using the pyrosequencing vacuum prep tool and incubated with 0.4 μM sequencing primer in annealing buffer (20 mM Tris-acetate, 2 mM MgAc2; pH 7.6). The reaction was incubated at 80 °C for 2 min and cooled down to room temperature for 20 min to allow sequencing primer annealing. The methylation levels at the target CpGs were evaluated by converting the resulting pyrograms to numerical values for peak heights and expressed as the average of all patients for a given CpG site analyzed.

**RNA expression data and analysis**. Transcriptome data, measured by RNA sequencing (RNAseq), were downloaded from TCGA-SKCM project[9] and normalized with DESeq package[55]. As with the DNA methylation data, the normalized RNAseq data was first filtered by MAD for the 100 most variable transcripts and then analysed by PLS-DA to assess the discriminative potential of the transcriptome between UV-mutant and non UV-mutant cutaneous melanomas. We used quanTiseq package[56] to estimate the fractions of ten immune cell types using the RNAseq from TCGA-SKCM project[9], comparing UV and non UV-mutant melanoma patients. Then Mann–Whitney U Test was performed to compare the two conditions as this non-parametric test is robust to outliers, which were detected in some data points of the various cell types.

**Gene ontology and pathway analysis**. Gene ontology and pathway analysis were performed using the Jensen Disease ontology and KEGG pathway databases available on Enrichr website[57,58] (https://maayanlab.cloud/Enrichr). Given that genes with larger numbers of probes are more likely to have significantly differentially methylated CpGs, potentially biasing gene set analysis, we implemented the *gometh* function of the missMethyl package[59] in R to adjust for the number of CpGs per gene, which ranges on the 450K array from 1 to 1299 CpGs.

**Cross-OMICs and integrative analysis**. Regarding eQTMs, we applied Pearson correlation between RNA expression and DNA methylation data of the 169 genes

that are common between BCH and TCGA DMRs, while limiting the analysis of a given gene to its constituent CpGs.

For the integrated methylome-transcriptome analysis, we filtered each of the 450K and RNAseq datasets by MAD and applied PLS-DA independently to each OMICs, as described in previous sections. Next, we applied sparse PLS-DA[60] that uses LASSO[61] penalization technique to select the 25 most informative CpGs and transcript probes in each dataset (Supplementary Data 23). We then applied an integrative analysis on the subsets of methylation and transcriptomic data, together with UV exposure outcome. We used the DIABLO method to gain a better understanding of the interplay between the different levels of data that are measured[62]. All these analyses were done using mixOmics R package[63].

For the prediction of multi-OMICs driver score, genome, transcriptome and DNA methylome data were downloaded from TCGA-SKCM (473 cases)[9]. Then, we calculated for each gene: [CNV: Copy Number Variation], being the number of tumors affected by a deep insertion (≥ +2) or a deep deletion (≤ −2), and this number was divided by the maximum CNV value obtained across the analysed genes in order to generate the CNV score; [MUT: Mutation], being the number of tumors affected by at least one single-nucleotide alteration, and this number was divided by the maximum MUT value obtained across the analysed genes in order to generate the MUT score; [EXP: Expression], being the number of UV-mutant cases presenting variations in RNA expression ($|logFC| > 2$) relative to non UV-mutant patients, and this number was divided by the maximum EXP value obtained across the analysed genes in order to generate the EXP score; and [METH: Methylation], being the number of UV-mutant cases presenting variations in methylation ($|delta\ betas| > 0.1$) relative to non UV-mutant patients, and this number was divided by the maximum METH value obtained across the analysed genes in order to generate the METH score. The driver score for each gene was then calculated as the sum of these four proportions, representing a derivation of our recently reported cancer driver score[17] by additionally including DNA methylation data on top of genomic and transcriptomic data.

**Power estimates**. The OMIC with the largest dimension (i.e., involving an agnostic approach with a multitude of statistical tests) would require the largest number of samples analysed to maintain a high statistical power. In this work, it would be the DNA methylome (~450,000 tests) followed by the transcriptome (~20,000 tests). Although WGS has a larger dimension than either, it is not being used agnostically in this work, but rather to screen for specific mutational signatures or mutated genes known to be genetically altered in melanoma. Accordingly, statistical power is estimated based on the methylome as such: the overall mean standard deviation (SD) of methylation probes in the BCH or TCGA data is 0.11 (for methylation values ranging 0-1). Given an effect size ≥10% methylation difference (a threshold used in our prioritization filters as reported in Supplementary Figs. 1a, b, 2a) and based on an alpha of 0.05, we will have >80% power with at least 20 exposed cases and 20 controls. Our sample size is larger and encompasses 21 UV-mutant cutaneous and 91 non UV-mutant cutaneous melanomas from BCH and TCGA. In addition to single-OMIC analysis, we performed integrative OMICs analysis, which can depict small effects shared between OMICs and not detected in the individual analyses and, hence, could be performed on smaller sample sizes than single-OMIC analysis. A recent study also proposes a joint power method for all OMICs being integrated[64]; however, we preferred to estimate the power based on the OMICs with the largest dimension as a more conservative approach. Statistical power was further enhanced by implementing dimension reduction and SVA approaches (as described in the 450K analysis), and the false positive likelihood was reduced by monitory and correcting for potential inflation, by adjusting for multiple-testing, and by replication of findings in two independent cohorts as well as by two different techniques (array- and pyrosequencing-based).

**Other statistical analyses**. Enrichment analyses was done using Chi-Square test or, when sample sizes were small, Fisher's exact test as proposed by R. For Kaplan–Meier melanoma-specific survival analyses, methylation data were dichotomized using the mean methylation level as cut-off, and log-rank testing was used to evaluate differences between curves[65]. The various plots in the manuscript were generated using ggplot2 package[66], except for the heatmaps, which were generated using Heatmap plus package. All analyses were performed on R. $P$ values ≤ 0.05 were considered statistically significant. Adjustment for multiple testing was based on FDR < 0.05.

**Systematic literature search**. We performed a systematic literature search on PubMed to select papers published until May 2021 that analyzed DNA methylome-wide data in clinical melanoma samples. To this end, we used the following syntax: ((melanoma) OR (melanomas)) AND ((Global DNA methylation) OR (methylome) OR (methylome-wide) OR (DNA methylation)). In total, we found 867 studies of which 20 (Supplementary Data 1) were included in our analysis since they covered methylome-wide profiling rather than targeted DNA methylation assays, and they were also conducted on clinical samples rather than cell lines or animal models.

**Reporting summary**. Further information on research design is available in the Nature Research Reporting Summary linked to this article.

## Data availability

The 450K data generated in this study have been deposited in the GE0 database under accession code GSE202750. The WGS data are available in the ICGC database [https://dcc.icgc.org/projects/SKCA-BR]. The 450K, RNAseq and WES data on melanoma samples from TCGA were downloaded from the GDC Legacy Portal [https://portal.gdc.cancer.gov/legacy-archive/search/f] and cBioPortal [https://www.cbioportal.org/datasets]. All other data is available within the Supplementary Data. Source data are provided with this paper.

## Code availability

Bioinformatics pipelines used in this study are available in [https://zenodo.org/record/6530343#.Ynuzki-tFTY][67] under the DOI number: https://doi.org/10.5281/zenodo.6530343.

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

## Acknowledgements

We thank all melanoma patients who contributed to the study. This work was supported by Public Ministry of Labor, Campinas (Research, Prevention, and Education of Occupational Cancer), Barretos Cancer Hospital and IARC. A.L.S.A.V. was supported by FAPESP (grant no 2016/15941-3 and 2017/09612-0). We thank the BCH biobank for DNA isolation and sample processing and storage. The initial development of the multi-OMICs driver score was partially supported by the grant from the Institut National du Cancer (INCa, France), La direction générale de l'offre de soins (DGOS), and INSERM (SIRIC LYriCAN, INCa-DGOS-Inserm_12563) to Z.H. Where authors are identified as personnel of the IARC/WHO, the authors alone are responsible for the views expressed in this article and they do not necessarily represent the decisions, policy or views of the IARC/WHO.

## Author contributions

Conceptualization: A.L.S.A.V., V.L.V., and A.G. Patient recruitment and clinical data collection in BCH: C.S.C., A.L.C., and V.L.V. Methylome array design: A.L.S.A.V. and A.G. Generation of array and pyrosequencing methylation data: C.C. Analysis of OMICs data: A.L.S.A.V., A.N., V.C., Z.A., N.S., A.F.E., and A.G. Data interpretation: A.L.S.A.V., Z.H., R.M.R., V.L.V., and A.G. Supervision: A.G., Z.H., and V.L.V. Writing original draft: A.L.S.A.V., V.L.V., and A.G. Editing and reviewing the paper: all authors.

## Competing interests

The authors declare no competing interests.
