## [Peer Review File · Nature Communications]

Cutaneous and acral melanoma cross-OMICs reveals prognostic cancer drivers associated with pathobiology and ultraviolet exposureREVIEWER COMMENTS

Reviewer #1 (Remarks to the Author): expertise in clinical statistics and epidemiology

In this paper, the authors conducted a multi-OMICS analysis in cutaneous and acral melanomas and uncovered novel cancer driver genes affecting patient prognosis and biological mechanisms and biomarkers underpinning intrinsic pathological characteristics and extrinsic responses to UV exposure. The goal of the study is straightforward. There are loads of data. But beyond raw results, little new knowledge or insights, seem to be produced from the study except for the TAPBP gene. Here are some specific questions.

I was under the impression that new data, RNA-seq, DNA methylation data are produced for the BCH cohort. But there is no mention that such data is made publicly available. This is unacceptable.

Figure 2B looks puzzling. It seems that there are much more hypomethylation and hypermethylation events occur in BCH cohort than in TCGA cohort. Is this correct? And why? Also, what is the purpose of showing these two plots. What message is trying to get across here?

Figure 3B. Why there is no BRAF in non UV-mutant?

Figure 3C. What is the point of this Figure? What biological insight is shown here?

Figure 3D. What is the number above each cell type? Are these p-value? If so, what is the point showing them? Why not just highlight the significant use "*"?

Figure 5B. I don't get the point. It seems meaningless and suggest deletion.

Figure 5C seems mislabeled as 5D in Figure legend,

Figure 6E seems meaningless. Don't see any point showing here.

Reviewer #2 (Remarks to the Author): expert in melanoma -omics

In this study, Silva Almeida Vicente and collaborators investigate the methylation landscape of cutaneous melanomas from two sources (a clinical centre in Barretos, Brazil) and TCGA. They are primarily interested in the comparison between samples whose mutational landscape is dominated by UV vs those that are not. They find large overlap in methylated regions in both cohorts, which provides credibility that the analysis is robust and reproducible. They find relationships with driver mutational status, and potential genes and mechanisms to explain differences between UV-driven and non-UV driven melanoma. One of the main results is that patients who have tumours dominated by a UV signature have better survival than those who do not in both cohorts, with one gene, TAPBP, being identified as a potential new driver. I truly believe this is a carefully done study, nicely presented, with appropriate controls and statistical considerations, with an exciting new result. I am also excited about the inclusion of non-European descent patients, who are so frequently forgotten in these kinds of studies. As such, I really want to congratulate the authors on this very nice piece of work, and in fact, it is the first time I review a paper where I have not identified any major issues. As such, all my suggestions are minor.

Suggestions/comments

1. Please specify the cosine similarity threshold that was used to determine if COSMIC signature 7 was present.

2. Line 156. "The resultant methylome map distinctly clustered UV-mutant from non UV-mutant patients in both BCH and TCGA (Figure 3A)." - I don't exactly agree with this assessment for the BCH cohort, the non UV mutant patients seem to be scattered in between the UV-mutant. Can the

authors clarify on what basis they are saying this?

3. Line 159, "1- BRAF, associated with younger patients and to BRAF and MITF" should this "to" be changed to "with"?

4. Line 184. "Specifically, dendritic and regulatory T cells were significantly more enriched in non UV-mutant than in UV-mutant cutaneous melanoma (Figure 3D)." - Can the test that was used to derive the p value be specified? It isn't clear to me if this difference is driven by the outlier that is appreciated in Figure 3D.

5. Line 227. "TAPBP and EIF2AK4 RNA expression levels did not significantly associate with patient survival (Figure S2B), suggesting that their methylation levels may be stronger prognostic markers than their transcript levels." - Do the authors have any idea why this may be, what other biological mechanisms TAPBP methylation (but not expression) may be reflecting or influencing?

6. Line 256. Could this title be simplified please? Perhaps it would be more easily understood like "The DNA methylome has sufficient discriminative potential of UV exposure status compared to the transcriptome, and resembles in non UV-mutant cutaneous melanoma that of acral melanoma"

7. Line 284. While reading this, I wonder - how does survival compare between the non-UV mutant vs UV mutant acral melanoma? Perhaps numbers are too small, but it would be very interesting to see whether the same is observed (i.e. UV-mutant have better survival).

8. Line 570. Can a couple of sentences be added about what PLS-DA is, so that readers unfamiliar with the term can follow the analyses?

9. Line 618. "P Values ≤ 0.05 were consider statistically significant." - this should be "considered"

10. It is suggested in the discussion that the observed association of TAPBP methylation levels with survival may be through its effects on transcription levels, however, how do authors explain that the transcriptional analysis did not show an association with survival? If TAPBP is being proposed as a target for epigenetic therapy, are authors suggesting that the relationship of TAPBP methylation and survival is causal, but perhaps is not acting through gene expression levels?

11. Line 387. The term Caucasian is outdated, and is now considered racist. Please replace it for "European-descent" or a similar term.

Reviewer #3 (Remarks to the Author): expert in cancer epigenetics

Vincente et al use a multi-OMIC approach (UV, methylome, transcriptional and mutational) to investigate the epigenetic differences between UV and non-UV related cutaneous and acral melanomas. UV-exposed patients have higher melanoma specific survival compared to non UV patients in the publicly available datasets from BCH and TCGA. There are a similar number of DMRs either hyper- or hypo-methylated that displayed enrichment in TSS, 5'UTR and CpG islands, as expected. Moreover, TN melanomas are enriched in non UV patients in both BCH and TCGA datasets whereas BRAF mutants are enriched in UV patients only in BCH data. Filtered DMRs identified various pathways related with the immune response, many of which were shared between both datasets. A moderate overlap of methylated regions from BCH and TCGA was observed, and there was a significant effect on survival when TAPBP was hypermethylated. As expected, TABP is also downregulated by RNA in the public datasets. It was further shown nicely that DNA methylation, rather than RNA expression, separated UV vs non UV samples using different methods, including LASSO.

Most of the data is correlative in nature and no mechanistic insights are provided into how any derived correlations impact melanoma biology. A large number of functional validation studies need to be done for any solid conclusions can be drawn. For example, there are no experiments regarding how DNA methylation (even at specific targets such as TAPBP) impacts melanoma. Some of the key experiments would be testing how blocking DNA methylation influences TAPBP

expression, and whether this can further influence proliferation, invasion, metastasis, etc. As such, TAPBP's role as a tumor suppressor/oncogene needs to be validated in multiple systems. Molecular mechanism of how DNA methylation change would influence other cellular processes would be required.

There are also major questions regarding the data presented as stated below.

Figure 2A: It is unclear if this is also based on methylation status? If not, this does not seem to be new information and should be performed based on methylation..

Figure 2B: While the hypermethylation in the TCGA is rather striking, the BCH data is not. Also, has this information been shown in any other previously published papers?

Figure 2C-D: One would expect to see a high percentage of DNA methylation enriched within promoters, TSS and CpG regions. This is good for confirmation, but does not seem like new information for a main figure.

Figure 3A: There is not any separation between mutational subtypes using this clustering method? Why were mutational subtypes further investigated?

Notably, there does not seem to be good separation between UV and non UV status using this method in the BCH dataset.

Figure 3B: It is unclear if this is also based on methylation status? If not, this does not seem to be new information and should be performed based on methylation.

Figure 3D: With RNA-seq data available, were immune pathways also identified when using all differentially expressed genes (i.e. GSEA).

Figure 4A: In both datasets there is a rather low number of overlapping genes and CpGs from DMRs. This is rather concerning with no functional validation.

Figure 5: Although it is interesting that TAPBP is downregulated based on increased methylation, however, it is very concerning there is no functional validation presented (i.e. disrupting the TAPBP promoter, epistasis experiments for TAPBP on proliferation, invasion, etc).

Minor: Figure 5D is not shown.

Figure 6D: It is unclear if this is also based on methylation status? If not, this does not seem to be new information and should be performed based on methylation.

Figure 6E: Were these pathways also identified in the TCGA cohort? This should be shown or referenced.

Reviewer #1 (Remarks to the Author): expertise in clinical statistics and epidemiology:

In this paper, the authors conducted a multi-OMICs analysis in cutaneous and acral melanomas and uncovered novel cancer driver genes affecting patient prognosis and biological mechanisms and biomarkers underpinning intrinsic pathological characteristics and extrinsic responses to UV exposure. The goal of the study is straightforward. There are loads of data. But beyond raw results, little new knowledge or insights, seem to be produced from the study except for the *TABPB* gene. Here are some specific questions.

We thank the reviewer for the assessment. Unless some information was not entirely clear, the loads of data generated and analyzed yielded a substantial amount of new knowledge not limited only to the findings of the *TABPB* gene (which *per se* were important as well). Specifically:

1-Epigenetic modifications induced by UV exposure have not been investigated yet in clinical biospecimen, and we demonstrate their translational potential. This is the first paper showing that UV exposure produces specific alterations in DNA methylation in two clinical cohorts of cutaneous melanoma and the first report describing them. DNA methylome, genome and transcriptome analyses of cutaneous melanoma identified UV-related alterations in immunological pathways and revealed novel genes with cancer driver potential and/or effect on patient survival. Among those genes (and besides *TAPBP*), the paper also describes other genes that can be further investigated in larger cohorts (e.g. *FIGNL2*, *IFNLR1* and *EIF2AK4*, to name a few).

2-This is the first epigenomics comparative study encompassing both cutaneous and acral melanoma types, including white and dark skin phenotypes. It characterized the DNA methylome landscape of both melanoma types and will put this wealth of data at the service of the scientific community for further exploratory analyses.

3-Points 1 and 2 were followed by an in-depth analysis of the functional and clinical implications of the epigenetic alterations as well as an integrative DNA methylome, genome and transcriptome analysis to cross-knit the molecular mechanisms affecting regulatory regions, biological pathways, gene transcription, cancer driver potential, tumor classification and patient survival.

4- The DNA methylome also elucidated the relative contributions of intrinsic pathological and extrinsic UV-related differences towards shaping the melanoma epigenome. Unsupervised epigenomic mapping demonstrated that non UV-mutant cutaneous melanoma more closely resembles acral rather than UV-exposed cutaneous melanoma. This highlights that, even though the UV-mutant and non UV-mutant cutaneous melanomas are supposed to share the same pathological/cellular origin, they may need to be classified separately based on their underlying epigenomic landscape, which has the potential to capture markers of both exposure and cell identity. Moreover, the non UV-mutant cutaneous melanoma, by resembling in its epigenome the acral melanoma, may have a poorer prognosis and require a different therapeutic approach than the UV-exposed cutaneous melanoma.

5- This study encompasses the first multi-OMICs data on Brazilian melanoma patients, a highly admixture genetic population and representative of the Latin-American community, and contributes to overcoming a timely and important topic about the lack and critical need of genomics data derived from populations of non-European descent. In fact, less than 5% of genetic studies worldwide include participants with multiple ancestry^{1,2}. Therefore, the current study helps in diminishing population disparities and reducing inequalities in knowledge on melanoma research. Additionally, we place the wealth of the data generated from this work at the service of the public scientific community.

References:

1. Alicea, G.M., Rebecca, V.W. Un-Fair Skin: racial disparities in acral melanoma research. Nat Rev Cancer 22, 127–128 (2022).
2. Faturno S et al. A roadmap to increase diversity in genomic studies. Nat Med, 28, 243-250(2022).

1- I was under the impression that new data, RNA-seq, DNA methylation data are produced for the BCH cohort. But there is no mention that such data is made publicly available. This is unacceptable.

At the time of the paper's submission, we had included a Data Availability Statement in the Nature Portfolio Smart Report stating that:

1) The 450K, RNA-seq and Whole Exome Sequencing data on melanoma samples from TCGA were downloaded from the GDC Legacy Portal (<https://portal.gdc.cancer.gov/legacy-archive/search/f>) and cBioPortal (<https://www.cbioportal.org/datasets>).

Reference publication: Cancer Genome Atlas, N. Genomic Classification of Cutaneous Melanoma. Cell 161, 1681-96 (2015).

2) Regarding the whole genome sequencing and 450K data that we generated from melanoma samples in the Barretos Cancer Hospital, we already deposited the whole genome sequencing data in the International Cancer Genome Consortium database (<https://dcc.icgc.org/projects/SKCA-BR>) and shall publicly deposit the remaining HM450 data in case the paper is accepted for publication.

We have now incorporated the section about Data Availability into the manuscript's body (page 29, lines 659-664).

2- Figure 2B looks puzzling. It seems that there are much more hypomethylation and hypermethylation events occur in BCH cohort than in TCGA cohort. Is this correct? And why? Also, what is the purpose of showing these two plots. What message is trying to get across here?

Figure 2B shows the chromosome location of probes that were hypomethylated or hypermethylated in our two cohorts when we compare patients who have the UV signature to those who do not. Indeed, the number of detectable signals was higher in our BCH relative to the TCGA cohort, and this could be due in the BCH dataset to (1) the better quality of samples and/or their processing using our in-house optimized automated workflow to generate DNA methylome data, coupled to a priori designed sample distribution on the array that minimizes confounding with batch effects based on statistical semi-randomization as explained on page 23, lines 491-498 and (2) more accurate technology, using whole genome sequencing rather than whole exome sequencing, to assess UV exposure as explained on page 21, lines 437-448, and page 17, lines 370-374. Still, the distribution of the detected signals was similar between the two cohorts, whether across genomic locations or in the direction of the effects (both studies had approximately 60% hypermethylation and 40% hypomethylation sites), as demonstrated in Figure 2B. This was reinforced by consistent findings in both cohorts at the CpG, gene and biological pathway levels. This information is also available on pages 15, lines 304-310. In the revised version, we have now better emphasized why we expect better quality signals in BCH relative to TCGA (please see pages 14-15, lines 298-304), and we thank the reviewer for this comment which reinforces the importance of our newly generated data.

Extracted from page 23, lines 491-498:

"For BCH, bisulfite converted DNA samples were profiled using high-density methylation array (HM450, Illumina Inc., San Diego, CA, USA) and a well-established workflow optimized at the IARC Epigenomics and Mechanisms Branch for high-throughput analyses through an automated robotic system (Freedom EVO 150 by Tecan, Mannedorf, Switzerland) that can process the chips with minimal human error. Chips are scanned using Illumina iScan to produce two-color raw data files (.idat format). Sample allocation to the arrays was based on a semi-randomization design that ensures minimum confounding by technical variation and minimizes the masking of biological covariates of interest by batch effects."

Extracted from page 21, lines 437-448:

“Patients were recruited at BCH in the context of ICGC-Brazil project (International Cancer Genome Consortium), which encompassed 100 melanoma patients prior to any systemic treatment and from whom paired tumor/blood was profiled by WGS³² and tumor tissues were analysed by 450K DNA methylation array. ... The second cohort comprised 58 cutaneous melanoma samples from TCGA-SKCM for which information about UV signature was available (Figure 1) based on WES.”

Extracted from page 17, lines 370-374:

“Our findings also corroborate those of another study showing that cutaneous melanoma patients harbouring the UV mutation signature had higher disease-free and overall survival⁴¹. The consistency between findings is notable especially that our dataset included a different ethnic group and a more accurate methodology (WGS rather than WES) to predict the UV signature.”

Extracted from page 15, lines 304-310:

“Even though there were some clinicopathological and ethnic dissimilarities (Table 1) and methodological differences between the two cohorts in predicting UV signature status (WGS versus WES, respectively), we observed consistent findings in both at the CpG, gene and biological pathway levels. Moreover, UV-related DNA methylation alterations showed similar distributions between BCH and TCGA in hypo- and hyper-methylated regions as well as similar enrichments in regulatory and CpG density regions, in skin disorders and in immunological pathways.”

Extracted from pages 14-15, lines 298-304:

“The number of detectable signals was higher in the BCH relative to the TCGA cohort, and this is probably not due to statistical power differences as both datasets had very similar sample sizes. This could be rather due in the BCH dataset to (1) the better quality of samples and/or their processing using our in-house optimized automated workflow to generate DNA methylome data, coupled to a priori designed sample distribution on the array that minimizes confounding with batch effects based on statistical semi-randomization, and (2) more accurate technology, using WGS rather than WES, to assess UV exposure.”

3- Figure 3B. Why there is no BRAF in non UV-mutant?

BRAF mutation has been shown to be associated with UV-exposure (specifically non-chronic sun-damage skin)^{15,16}. Our study reinforces these findings, which also serve as independent positive controls for the high-quality and reproducible potential of our data. Still, had the sample sizes of non UV-mutant melanomas been much larger, some *BRAF* mutation might have been incidentally observed. We clarified this information within the pertinent section on page 7, lines 122-124:

“We also observed that the BRAF mutant group was the most enriched in UV-mutant patients in both BCH and TCGA, though reaching statistical significance only in BCH (Figures 3a-b). This was in line with other studies^{15,16}, further reinforcing the reproducibility potential of our data.”

References:

15. Kim, SY et al. Metaanalysis of BRAF mutations and clinicopathologic characteristics in primary melanoma. J Am acad Dermatol 72, 1036-1046(2015).
16. Lee, JH et al. Frequencies of BRAF and NRAS mutations are different in histological types and sites of origin of cutaneous melanoma: a meta-analysis. Br j Dermatol 164, 776-784(2011).

4- Figure 3C. What is the point of this Figure? What biological insight is shown here?

Figure 3c shows the epigenetically deregulated biological pathways in melanoma in response to UV exposure. We have observed that UV exposure induces DNA methylation alterations affecting genes with immune

system function and pathways related to autoimmune disease development. These observations motivated our in-depth investigation of differences in immune cell compositions in Figure 3d. The specific genes, their effect sizes, their directions of effect, the pathways they constitute and the interactions between the genes/pathways are described as well in the figure and on page 7 line 129-142 as such:

“Jensen disease analysis of the filtered DMRs showed a significant implication of the differentially methylated genes in skin disorders, such as systemic sclerosis (BCH and TCGA), vitiligo (BCH and TCGA), melanoma (BCH), and skin cancer (BCH), particularly among the top and false discovery rate (FDR)-adjusted ontologies (FDR < 0.05) (Supplementary Data 9-10). A number of other cancers and diseases were significantly enriched as well (Supplementary Data 9-10). This was complemented by KEGG pathway analysis, revealing 28 and 30 significant pathways ($p < 0.05$) in BCH and TCGA, respectively. Among them, a large proportion (10 pathways) were identical between BCH and TCGA, 8 and 6 of which remained significant after adjustment for the number of CpGs per gene and FDR, respectively (Figure 3c, Supplementary Data 11-14). These pathways constituted of differentially methylated genes implicated in immune system regulation: hematopoietic cell lineage, allograft rejection, graft-versus-host disease, intestinal immune network for IgA production, antigen processing presentation, inflammatory bowel disease, and relatedly, autoimmune diseases, such as type 1 diabetes mellitus, autoimmune thyroid disease, systemic lupus erythematosus and rheumatoid arthritis (Figure 3c).”

- 5- Figure 3D. What is the number above each cell type? Are these p-value? If so, what is the point showing them? Why not just highlight the significant use “*”?

Indeed, these numbers are p-values showing the statistical differences between the two groups, UV-mutant and non UV-mutant, for each immune cell type quantity. We have now indicated the significant values with stars as suggested by the reviewer, and this was explained in the legend of Figure 3d as such:

*“d) Immune cell composition inferred from RNA-sequencing data comparing UV-mutant and non UV-mutant cutaneous melanoma patients from TCGA. * $p < 0.05$, by Mann-Whitney U Test.”*

- 6- Figure 5B. I don't get the point. It seems meaningless and suggest deletion.

Figure 5b describes the cancer driver potential of the genes that are epigenetically altered by UV exposure in both the BCH and TCGA datasets and the genes for which methylation levels functionally impacted their transcription. The cancer driver potential was predicted using our recent multi-OMICs driver score¹⁷, by integrating copy number variation, point mutations, RNA expression and DNA methylation data profiled in melanoma patients. Importantly, this led to the identification of genes that have cancer driver potential in melanomagenesis, and those genes were compared using the same score to other genes known to be melanoma drivers (as assessed by other methods). Our results validated the previously known driver genes in melanoma, using an independent method (being our multi-OMICs driver score), and this equally served as a positive control to validate our method which predicted new genes with driver roles. This was described on page 10, lines 214-230 as such:

“Next, we pooled all 36 CpGs prioritized in Supplementary Fig. 1b (being common between BCH and TCGA) with the 10 CpGs prioritized in Supplementary Fig. 2a (being significant eQTM) and investigated their cancer driver potential derived from our recent multi-OMICs driver score¹⁷. This was performed using data on copy number variation, point mutations, RNA expression and DNA methylation profiled in cutaneous melanoma patients. We found that the top half of the CpGs with the highest cancer driver potential were largely predominated by CpGs of the TAPBP gene (Figure 5b) and that this gene ranked among the top 4 driver genes when methylation levels were averaged across CpGs of a given gene (Supplementary Fig. 2c).

As a positive control, we used a list of genes known to play a driver role in cutaneous melanoma based on the ConsensusDriver score method (i.e. with ConsensusDriver > 1.5)³, which preferentially selects cancer driver genes

that are frequently mutated in tumor tissues. We calculated the multi-OMICs driver scores for those genes, derived by measuring the extent of their OMICs alterations in UV-mutant relative to non UV-mutant melanomas (Supplementary Fig. 2d), as was done for the experimental gene set (Figure 5b). We found that the multi-OMICs driver scores of the latter, including TAPBP, were predominantly in the same range as that of the positive control genes (1.24 - 2.50) (Supplementary Fig. 2d), reinforcing the cancer driver potential of the experimental gene set relative to known driver genes in melanoma.”

References:

17. Halaburkova, A. et al. Pan-cancer multi-omics analysis and orthogonal experimental assessment of epigenetic driver genes. *Genome Res* 30, 1517-1532 (2020).

6- Figure 5C seems mislabeled as 5D in Figure legend,

Thank you for noting this. It was corrected accordingly.

7- Figure 6E seems meaningless. Don't see any point showing here.

Figure 6e describes the genes and pathways epigenetically deregulated between cutaneous and acral melanomas while focusing on their non UV-mutant forms in order to eliminate the effects of UV exposure (and, hence, focus this part on pathologically related epigenetic alterations). This complements the results of previous sections in the manuscript which focused on UV-related epigenetic effects in melanoma, allows comparative evaluation of UV- *versus* pathology-related epigenetic effects, and provides a better understanding of epigenetic mechanisms contributing to pathological differences between cutaneous and acral melanomas, especially that this has not been described in other studies. Even though the pertinent biological pathways in Figure 6E did not pass FDR significance, it is important to show these results because (1) the analysis should not be biased towards reporting only strongly (FDR) significant findings, and (2) the lack of FDR significance supports our finding that the extrinsic effect of UV on epigenetic alterations (which were FDR significant) is stronger than the intrinsic effect of pathology in dictating melanoma subtypes.

Reviewer #2 (Remarks to the Author): expert in melanoma -omics:

In this study, Silva Almeida Vicente and collaborators investigate the methylation landscape of cutaneous melanomas from two sources (a clinical centre in Barretos, Brazil) and TCGA. They are primarily interested in the comparison between samples whose mutational landscape is dominated by UV vs those that are not. They find large overlap in methylated regions in both cohorts, which provides credibility that the analysis is robust and reproducible. They find relationships with driver mutational status, and potential genes and mechanisms to explain differences between UV-driven and non-UV driven melanoma. One of the main results is that patients who have tumours dominated by a UV signature have better survival than those who do not in both cohorts, with one gene, TAPBP, being identified as a potential new driver. I truly believe this is a carefully done study, nicely presented, with appropriate controls and statistical considerations, with an exciting new result. I am also excited about the inclusion of non-European descent patients, who are so frequently forgotten in these kinds of studies. As such, I really want to congratulate the authors on this very nice piece of work, and in fact, it is the first time I review a paper where I have not identified any major issues. As such, all my suggestions are minor.

We thank the reviewer for the assessment of this work, including the encouraging words. Even though the reviewer considers all his/her suggestions as minor, we felt they were important and have addressed them in detail below. Some of these comments led to substantial changes and improvements to the manuscript.

Suggestions/comments

1- Please specify the cosine similarity threshold that was used to determine if COSMIC signature 7 was present.

We used the Non-negative Matrix Factorization (NMF) algorithm³⁹ to determine the consensus signatures among the 71 patients and associate with the COSMIC signature 7. At the moment of the analyses, we used the 21 signatures¹⁰ that were available and identified a consensus signature with more than 0.8 cosine similarity. This information is now available in the Methods section on page 22, lines 474-481 as such:

“The UV mutational signature identification was performed using the SomaticSignatures Bioconductor package³⁸. We used the Non-negative Matrix Factorization (NMF) algorithm³⁹ to determine the consensus signatures among the 71 patients and associate with the COSMIC signature 7. At the moment of the analyses, we used the 21 signatures¹⁰ that were available and identified a consensus signature with more than 0.8 cosine similarity. For both BCH and TCGA cohorts, we classified samples as harboring an UV mutation signature (Cosmic Signature 7) based on the recommended criteria in which C>T transitions at dipyrimidine sites accounted for more than 60% or CC>TT mutations more than 5% of the total mutation burden⁹.”

References:

10. Alexandrov LB et al. Signatures of mutational processes in human cancer. *Nature*, 500, 415-421(2013).
39. Lee DD et al. Learning the parts of objects by non-negative matrix factorization. *Nature*, 6755, 788-791(1999).

2. Line 156. “The resultant methylome map distinctly clustered UV-mutant from non UV-mutant patients in both BCH and TCGA (Figure 3A).” – I don’t exactly agree with this assessment for the BCH cohort, the non UV mutant patients seem to be scattered in between the UV-mutant. Can the authors clarify on what basis they are saying this?

The hierarchical cluster was based on Euclidean distance. In either dataset, the statistics did produce one cluster fully occupied by non UV-mutant samples (being the rate limiting exposure group); this cluster is termed C1 in BCH and C3 in TCGA, as defined by Euclidean distance (**New Figure 3a**). Even if we increase the stringency by grouping adjacent clusters to increase the cluster size, the grouped cluster remains enriched in non UV-mutants. Few non relevant samples may expectedly fall in or out of a given cluster due to residual data variability that persist or can confound the UV methylation signatures in some samples. To better quantify this overall assessment, we performed enrichment analysis for those clusters using chi-square test while delimiting the cluster boundaries by the limits statistically specified by Euclidean distance. In the BCH cohort, cluster C1 was fully occupied by non UV-mutant samples (**New Figure 3a**), and its DNA methylation profile was also visually distinct from the other clusters, exhibiting a

New Figure 3a. This is an exact copy of the previously submitted Figure 3a but adding cluster annotations on top of the heatmap.

clear hypermethylation (red) and hypomethylation (blue) in the upper and lower panels of its CpGs, respectively, relative to the other clusters. Even if we merge C1 with the adjacent cluster C2, the non UV-mutant patients remain statistically enriched in this combined cluster (p -value= $1.95e-03$), which now encompasses almost all non UV-mutants analyzed. The same approach was used in TCGA cohort. Cluster C3 was fully occupied by non UV-mutant samples and was visually distinct from the other clusters, exhibiting again a clear hypermethylation (red) and hypomethylation (blue) in the upper and lower panels of its CpGs, respectively, relative to the other clusters. Even if we merge C3 with the adjacent cluster C4, the non UV-mutant patients remain statistically enriched in this combined cluster (p -value= $3.00e-10$), which now encompasses almost all non UV-mutants analyzed (**New Figure 3a**). We thank the reviewer for this comment which strengthened this result, and we have added this information into the revised manuscript on page 6, line 102-114 as such:

“The resultant methylome map distinctly clustered UV-mutant from non UV-mutant patients in both BCH and TCGA (Figure 3a). In the BCH cohort, cluster C1 (as defined by Euclidean distance) was fully occupied by non UV-mutant samples (Figure 3a) and exhibited a DNA methylation profile that was visually distinct, with an upper hypermethylation (red) stretch and a lower hypomethylation (blue) stretch, relative to the other clusters. Even if C1 is merged with the adjacent cluster C2, the non UV-mutant patients remain statistically enriched in this combined cluster (p -value= $1.95e-03$), which now encompasses almost all non UV-mutants analyzed. A similar pattern was observed in the TCGA cohort. Cluster C3 was fully occupied by non UV-mutant samples and was visually distinct, exhibiting again an upper hypermethylation (red) stretch and a lower hypomethylation (blue) stretch, relative to the other clusters. Even if C3 is merged with the adjacent cluster C4, the non UV-mutant patients remain statistically enriched in this combined cluster (p -value= $3.00e-10$), which now encompasses almost all non UV-mutants analyzed (Figure 3a).”

We also clarified this information in the legend of Figure 3a as such:

“a) Hierarchical clustering of cutaneous melanoma patients in BCH and TCGA based on methylation levels of 4,721 and 793 CpGs, respectively, as derived from Supplementary Fig. 1a. Enrichment analysis for non UV-mutant patients in clusters C1-C4 was performed using chi-square test while delimiting the cluster boundaries by the limits statistically specified by Euclidean distance.”

3. Line 159, “1- BRAF, associated with younger patients and to BRAF and MITF” should this “to” be changed to “with”?

Thank you for noting this. It was corrected accordingly.

4. Line 184. “Specifically, dendritic and regulatory T cells were significantly more enriched in non UV-mutant than in UV-mutant cutaneous melanoma (Figure 3D).” – Can the test that was used to derive the p value be specified? It isn't clear to me if this difference is driven by the outlier that is appreciated in Figure 3D.

Thank you for noting the outlier. We repeated the analysis using Mann-Whitney U Test as it is a non-parametric test robust to the outliers. We performed this test on all the immune cells. The Dendritic cells remain significant ($p = 0.03$) while the Regulatory T cells not. This information was incorporated into the main text (page 7, lines 143-148) as such:

“The role of DNA methylation alterations in regulating immune system function was investigated in further depth and validated using RNA sequencing data (Methods section), demonstrating that immune cell composition was indeed different between UV-mutant and non UV-mutant cutaneous melanoma patients (Figure 3D). Specifically, dendritic cells were significantly more infiltrated in the non UV-mutant than in UV-mutant cutaneous melanoma (Figure 3d, $p = 0.03$).

We also clarified this information in the legend of Figure 3d as such:

*“d) Immune cell composition inferred from RNA-sequencing data comparing UV-mutant and non UV-mutant cutaneous melanoma patients from TCGA. *p < 0.05, by Mann-Whitney U Test”.*

Similarly, we clarified this in the Methods on page 26, lines 579-583 as such:

*“We used `quanTiseq` package⁵² to estimate the fractions of ten immune cell types using the RNAseq from TCGA-SKCM project⁹, comparing UV and non UV-mutant melanoma patients. Then *Mann-Whitney U Test* was performed to compare the two conditions as this non-parametric test is robust to outliers, which were detected in some data points of the various cell types.”*

5. Line 227. “TAPBP and EIF2AK4 RNA expression levels did not significantly associate with patient survival (Figure S2B), suggesting that their methylation levels may be stronger prognostic markers than their transcript levels.” – Do the authors have any idea why this may be, what other biological mechanisms TAPBP methylation (but not expression) may be reflecting or influencing?

Even though the sample size of expression data was the same as that of methylation data, the association between expression and survival was not as significant as that between methylation and survival. This suggests that the difference is not merely due to statistical power but could indeed reflect biological basis. DNA methylation does not act solely through affecting gene transcription but is known to also associate with chromosomal instability, the induction of splice variants, alterations in enhancer regions, changes in microRNA binding regions and expression control regions, and mutations^{1,2,3}. Hence, DNA methylation may function as a prognostic marker *per se* or through these various non expression-related mechanisms. Our observation is in line with a multitude of studies highlighting the high sensitivity of the epigenome to exposure and risk factors^{4,5,6}. We have added this information in the Discussion on page 16, lines 341-350 as such:

*“TAPBP methylation significantly predicted patient prognosis in both BCH and TCGA. Even though the sample size of expression data was the same as that of methylation data, the association between TAPBP expression and survival was not significant. This suggests that the difference is not merely due to statistical power but could indeed reflect biological basis. DNA methylation does not act solely through affecting gene transcription but is known to also associate with chromosomal instability, the induction of splice variants, alterations in enhancer regions, changes in microRNA binding regions and expression control regions, and mutations²³⁻²⁵. Hence, DNA methylation may function as a prognostic marker *per se* or through these various non expression-related mechanisms. Our observation is in line with a multitude of studies highlighting the high sensitivity of the epigenome to exposure and risk factors^{12,26}.”*

References:

12. Herceg, Z. et al. Roadmap for investigating epigenome deregulation and environmental origins of cancer. *Int J Cancer* 142, 874-882 (2018).
23. Narayanan SP et al. A saga of cancer epigenetics: linking epigenetics to alternative splicing. *Biochem J*, 474, 885-896(2017).
24. Cho JW et al. The importance of enhancer methylation for epigenetic regulation of tumorigenesis in squamous lung cancer. *Exp Mol Med*, 54, 12-22(2022).
25. Anwar SL et al. DNA methylation, microRNAs, and their crosstalk as potential biomarkers in hepatocellular carcinoma. *World J Gastroenterol*, 24, 7894-7913(2014).
26. Bowers EC et al. Linking the epigenome with exposure effects and susceptibility: the epigenetic seed and soil model. *Toxicol Sci*, 155, 302-314(2017).

6. Line 256. Could this title be simplified please? Perhaps it would be more easily understood like "The DNA methylome has sufficient discriminative potential of UV exposure status compared to the transcriptome, and resembles in non UV-mutant cutaneous melanoma that of acral melanoma"

Thank you for noting this. The word count of the title had to be also decreased as per the journal's guidelines. It was corrected accordingly as such:

"Cutaneous and acral melanoma cross-OMICs: UV versus pathobiology"

7. Line 284. While reading this, I wonder - how does survival compare between the non-UV mutant vs UV mutant acral melanoma? Perhaps numbers are too small, but it would be very interesting to see whether the same is observed (i.e. UV-mutant have better survival).

We included the melanoma-specific survival of UV-mutant acral melanomas as suggested by the reviewer (please, see the figure to the right). The 3 UV-mutant acral patients presented a melanoma-specific survival in between (or similar to) the non-UV mutant acral and UV-mutant cutaneous melanoma patients, and a better survival than the non UV-mutant cutaneous melanoma patients. Although these results seem promising and in line with expectations, the very small sample size of UV-mutant acral patients does not allow a statistical evaluation, and the results cannot be interpreted with confidence. Accordingly, we decided not to include this version of the figure in the revised manuscript, but these results encouragingly call for future exploratory work with larger sample sizes.

8. Line 570. Can a couple of sentences be added about what PLS-DA is, so that readers unfamiliar with the term can follow the analyses?

Given of the loads of information in this manuscript and to eliminate redundancy, we had not described PLS-DA in the Results but had rather placed this information in the Methods on page 25, lines 549-557:

"In addition to generating DMPs and DMRs, methylation data from BCH and TCGA samples were further investigated using Partial Least Squares Discriminant Analysis (PLS-DA)¹⁶. This approach performs classification of samples using partial least squares regression of the categorical outcome Y (cancer subtype) on the predictor variables (DNA methylation). PLS-DA is a clustering technique that allows the quantification of the discrimination relevance of a given variable (CpG) and to predict the phenotype of new samples (independent of DMPs or DMRs). This method is especially suited to deal with a much larger number of variables than samples, as in next-generation microarray and sequencing data, and we aided this method further by a filtering step using median absolute deviation (MAD)¹⁷."

9. Line 618. "P Values ≤ 0.05 were consider statistically significant." - this should be "considered"

Thank you for noting this. It was corrected accordingly.

10. It is suggested in the discussion that the observed association of TAPBP methylation levels with survival may be through its effects on transcription levels, however, how do authors explain that the transcriptional analysis did not show an association with survival? If TAPBP is being proposed as a target for epigenetic therapy, are authors suggesting that the relationship of TAPBP methylation and survival is causal, but perhaps is not acting through gene expression levels?

We have addressed the part on expression in point 5 of your comments and explained in the Discussion that DNA methylation may function through other mechanisms besides transcription, such as affecting chromosomal instability, the induction of splice variants, alterations in enhancer regions, etc... Regarding the aspect on causality, the relation between *TAPBP* methylation and survival may not necessarily be causal. Our results, however, pinpoint to an increased likelihood of causality because (1) they were reproduced in two independent populations, including different ethnicities, which offer a natural means of effect randomization (hence, minimizing the influence of confounders), (2) they showed a dose-response (*TAPBP* hypomethylation was associated with increased survival relative to hypermethylation), and (3) they yielded a cancer driver potential for *TAPBP* that was comparable to that of known cancer driver genes. Still, more analysis and datasets will be needed to better reinforce the causality of the associations (for example, by using in larger sample sizes germline data as proxies for *TAPBP* methylation through Mendelian Randomization, but which requires seeking data access and ethical approvals and a substantial amount of funding and analysis that are beyond the timeframe and resources of the review process). We have now described the likelihood of causality of our findings as well as how to reinforce them in the discussion on page 16, lines 351-359 as such:

"The relation between TAPBP methylation and survival may not be necessarily causal. Our results, however, pinpoint to an increased likelihood of causality because (1) they were reproduced in two independent populations, including different ethnicities, which offer a natural means of effect randomization (hence, minimizing the influence of confounders), (2) they showed a dose-response (TAPBP hypomethylation was associated with increased survival relative to hypermethylation), and (3) they yielded a cancer driver potential for TAPBP that was comparable to that of known cancer driver genes. Still, more datasets will be needed to better reinforce the causality of the associations, for example, by using in larger sample sizes germline data as proxies for TAPBP methylation through Mendelian Randomization."

11. Line 387. The term Caucasian is outdated, and is now considered racist. Please replace it for "European-descent" or a similar term.

Thank you for noting this. It was corrected accordingly.

Reviewer #3 (Remarks to the Author): expert in cancer epigenetics

Vincente et al use a multi-OMIC approach (UV, methylome, transcriptional and mutational) to investigate the epigenetic differences between UV and non-UV related cutaneous and acral melanomas. UV-exposed patients have higher melanoma specific survival compared to non UV patients in the publicly available datasets from BCH and TCGA. There are a similar number of DMRs either hyper- or hypo-methylated that displayed enrichment in TSS, 5'UTR and CpG islands, as expected. Moreover, TN melanomas are enriched in non UV patients in both BCH and TCGA datasets whereas BRAF mutants are enriched in UV patients only in BCH data. Filtered DMRs identified various pathways related with the immune response, many of which were shared between both datasets. A moderate overlap of methylated regions from BCH and TCGA was observed, and there was a significant effect on survival when TAPBP was hypermethylated. As expected, TABP is also downregulated by RNA in the public datasets. It was further shown nicely that DNA methylation, rather than RNA expression, separated UV vs non UV samples using different methods, including LASSO.

Most of the data is correlative in nature and no mechanistic insights are provided into how any derived correlations impact melanoma biology. A large number of functional validation studies need to be done for any solid conclusions can be drawn. For example, there are no experiments regarding how DNA methylation (even at specific targets such as TAPBP) impacts melanoma. Some of the key experiments would be testing how blocking DNA methylation influences TAPBP expression, and whether this can further influence proliferation, invasion, metastasis, etc. As such, TAPBP's role as a tumor suppressor/oncogene needs to be validated in multiple systems. Molecular mechanism of how DNA methylation change would influence other cellular processes would be required.

We thank the reviewer for this assessment and agree about the importance of further mechanistic studies using multiple systems. It is also important to keep in mind, however, that such further aims need to be examined in the same system to be directly comparable to the current findings. That is, since the findings of this work have been investigated in human clinical samples, suggested experiments (such as blocking DNA methylation to assess its effect on TAPBP expression) would need to be investigated in human clinical samples as well, such as using samples derived from intervention studies that are beyond the reach of the current work. Such experiments can be more easily investigated in cell line or mouse models but would not be directly comparable to the system used in this work. Having experience in both mechanistic *in vitro* and human clinical models in our team, we and others prefer to view these models as being complementary rather than directly comparable. Hence, the mechanistic aspect of this work may better fit in subsequent independent studies that build on our findings, and we agree with the reviewer about its importance.

There are also major questions regarding the data presented as stated below.

1. Figure 2A: It is unclear if this is also based on methylation status? If not, this does not seem to be new information and should be performed based on methylation.

We had specified in the manuscript that only one study showed that patients harboring the UV mutation signature presented longer disease-free and overall survival¹¹. However, Figure 2a shows for the first time that patients harboring the UV mutation signature presented higher melanoma-specific survival, using a more accurate approach for predicting UV signature status and new ethnicity group (TCGA predicted it by using whole exome sequencing, whereas in BCH we used whole genome sequencing). These results also serve as positive controls for the high-quality and reproducible potential of our data. We have further clarified that in text on page 17, lines 370-374 as such:

"Our findings also corroborate those of another study showing that cutaneous melanoma patients harbouring the UV mutation signature had higher disease-free and overall survival¹¹. The consistency between findings is notable especially that our dataset included a different ethnic group and a more accurate methodology (WGS rather than WES) to predict the UV signature."

References:

11. Trucco, L.D. et al. Ultraviolet radiation-induced DNA damage is prognostic for outcome in melanoma. *Nat Med* **25**, 221-224 (2019).

2. Figure 2B: While the hypermethylation in the TCGA is rather striking, the BCH data is not. Also, has this information been shown in any other previously published papers?

The distribution of the detected methylation signals were similar between the two cohorts, whether across genomic locations or in the direction of the effects (both studies had approximately 60% hypermethylation and 40% hypomethylation sites), as demonstrated in Figure 2b. This was reinforced by consistent findings in both

cohorts at the CpG, gene and biological pathway levels. Our study is the first to report DNA methylation changes in melanoma tissues related to UV exposure, the major risk factor for melanoma. This point raised by the reviewer has been also addressed in further detail under point 2 of Reviewer #1.

3. Figure 2C-D: One would expect to see a high percentage of DNA methylation enriched within promoters, TSS and CpG regions. This is good for confirmation, but does not seem like new information for a main figure.

As we explained in our response to the previous comment, our study is the first to report DNA methylation changes related to UV exposure in melanoma samples. It is also important in the early figure panels (2c-d) of the manuscript to provide a descriptive overview of the distribution of methylation changes across chromosomal regions, CpG density regions and regulatory elements, stratified by the direction of effect (hypo- versus hyper-methylation). We simplified the text to better highlight the major findings of this figure, as evident on page 5, lines 86-95:

“In BCH melanomas, of the 2,620 DMRs, 1,541 (58.8%) were hypermethylated and 1,079 (41.2%) were hypomethylated (Figure 2b; Supplementary Data 2). A similar proportion of hypermethylated (62.8%; 378 out of 602) and hypomethylated (37.2%; 224 out of 602) DMRs was observed in TCGA (Figure 2b; Supplementary Data 6). The enrichment distributions in CpG regulatory or density regions were also similar in both cohorts. Specifically, in CpG regulatory regions, the significant enrichments in both cohorts were those of hypomethylated DMRs in regions 1-5Kb upstream of the transcription start site and of hypermethylated DMRs in promoters, exon/intron boundaries and 5’UTR ($p < 0.001$) (Figure 2c). In CpG density regions, the significant enrichments were those of hyper- or hypomethylated DMRs in CpG islands or shores ($p < 0.001$) (Figure 2d).”

4. Figure 3A: There is not any separation between mutational subtypes using this clustering method? Why were mutational subtypes further investigated? Notably, there does not seem to be good separation between UV and non UV status using this method in the BCH dataset.

We thank the reviewer for this valuable comment, which is similar to point 2 raised by Reviewer #2. In addition to our detailed response to corresponding point 2 of Reviewer #2 (also elaborated further in our response to point 3 of Reviewer #1), we add that mutational subtypes represent a well-established approach to classify cutaneous melanoma^{1,2}. Our study reinforces these findings, which also serve as independent positive controls for the high-quality and reproducible potential of our data. Our findings also show that UV exposure produces DNA methylation changes in genes that can be different from critical ones mutationally altered by the same environmental exposure (*BRAF*, *NF1* and *RAS* were not significantly differentially methylated in melanoma tissues in relation to UV exposure). We have added this information on page 7, lines 123-128 as such:

“This was in line with other studies^{15,16}, further reinforcing the reproducibility potential of our data. Interestingly, BRAF, NF1 and RAS were not significantly differentially methylated in melanoma tissues in relation to UV exposure (Supplementary Data 7-8), highlighting that UV exposure produces DNA methylation changes in genes that can be different from critical ones mutationally altered by the same environmental exposure.”

References:

1. Cancer Genome Atlas Network. Genomic Classification of Cutaneous Melanoma. Cell, 161, 1681-1696(2015).
2. Hayward et al. Whole-genome landscapes of major melanoma subtypes. Nature, 7653, 175-180(2017).
15. Kim, SY et al. Metaanalysis of BRAF mutations and clinicopathologic characteristics in primary melanoma. J Am acad Dermatol 72, 1036-1046(2015).

16.Lee, JH et al. Frequencies of BRAF and NRAS mutations are different in histological types and sites of origin of cutaneous melanoma: a meta-analysis. Br j Dermatol 164, 776-784(2011).

5. Figure 3B: It is unclear if this is also based on methylation status? If not, this does not seem to be new information and should be performed based on methylation.

Indeed, the molecular subtypes in Figure 3b are not analyzed in relation to methylation status but rather included to support, with statistical significance, their association with UV and methylation status described in Figure 3a. We elaborated further on this in point 4 above (including under point 3 to Reviewer #1 and under point 2 to Reviewer #2).

6. Figure 3D: With RNA-seq data available, were immune pathways also identified when using all differentially expressed genes (i.e. GSEA).

We do not necessarily expect that differentially expressed genes would be also enriched in immune pathways as differentially methylation genes, even though there could be a correlation between the two. The purpose for which we used expression data was to actually look whether there was infiltration of immune cells into the melanoma samples, given that we had observed that differentially methylated genes in these melanomas were enriched in immune pathways. Indeed, we found that dendritic cells were significantly infiltrated in the non UV-mutant relative to the UV-mutant melanomas. We clarified that in the Results section by adding the term 'infiltration' (page 8, line 147). Nevertheless, we did analyze differential expression between UV-mutant and Non UV-mutant melanomas based on the reviewer's comment as we think this could be valuable complementary information. We obtained the following results:

- Jensen Disease: several diseases were significant at nominal p-value ($p < 0.05$), including skin cancer and melanoma (Supplementary Data 16), and the second top most disease was Grave's disease ($FDR < 0.05$), an autoimmune disorder.
- KEGG pathways: several pathways were significant at nominal p-value ($p < 0.05$), including basal cell carcinoma, and only one pathway was FDR significant (Salivary secretion, $FDR = 0.040$, Supplementary Data 17).

Overall, we did see some enrichment in immune disorders and skin-related diseases in the differentially expressed genes, but we refrain to directly compare that to the differentially methylated pathways for the reasons described above and we rather reported them as complementary information on page 7, lines 143-151. We thank the reviewer for this comment which strengthened our findings.

Extracted from page 7, lines 143-151:

"The role of DNA methylation alterations in regulating immune system function was investigated in further depth and validated using RNA sequencing data (Methods section), demonstrating that immune cell composition was indeed different between UV-mutant and non UV-mutant cutaneous melanoma patients (Figure 3d). Specifically, dendritic cells were significantly more infiltrated in the non UV-mutant than in UV-mutant cutaneous melanoma (Figure 3d, $p = 0.03$). Complementary analysis using differentially expressed genes comparing non UV-mutant and UV-mutant cutaneous melanoma patients ($p < 0.05$, Supplementary Data 15) also showed enrichment in immune disorders and skin-related diseases, though none reached FDR significance (Supplementary Data 16-17)."

7. Figure 4A: In both datasets there is a rather low number of overlapping genes and CpGs from DMRs. This is rather concerning with no functional validation.

We thank the reviewer for this comment. We would however like to reiterate the high level of replication we have observed between the two analyzed cohorts, including:

- 1- Similar proportions of hypermethylated (60%) and hypomethylated (40%) DMRs;
- 2- Similar enrichment in CpG regulatory regions, specifically enrichment of hypomethylated DMRs in regions 1-5Kb upstream of the transcription start site and of hypermethylated DMRs in promoters, exon/intron boundaries and 5'UTR.
- 3- Similar enrichment in CpG density regions, specifically enrichment of hyper- or hypo-methylated DMRs in CpG islands or shores.
- 4- A large portion of epigenetically deregulated pathways in common (10 identical pathways out of 28 and 30 pathways in BCH and TCGA, respectively);
- 5- Reproducible effects of *TAPBP* methylation on patient prognosis;
- 6- And even when we zoomed to single CpG resolution, the enrichment remained statistically significant between the two cohorts.
- 7- We also validated the top marker (*TAPBP* methylation) by an independent technique (pyrosequencing).

One also needs to keep in mind that there are important methodological, clinicopathological and ethnic differences between the two datasets as mentioned on pages 14-15, lines 297-310, and we focused the overlap only on CpGs with high effect size (> 10%) and having the same direction of effect between the two cohorts. If we reduce the stringency, the overlap increases.

Extracted from pages 14-15, lines 297-310:

"The only available melanoma dataset with methylome and genome data for replication of our BCH findings was from the TCGA. The number of detectable signals was higher in the BCH relative to the TCGA cohort, and this is probably not due to statistical power differences as both datasets had very similar sample sizes. This could be rather due in the BCH dataset to (1) the better quality of samples and/or their processing using our in-house optimized automated workflow to generate DNA methylome data, coupled to a priori designed sample distribution on the array that minimizes confounding with batch effects based on statistical semi-randomization, and (2) more accurate technology, using WGS rather than WES, to assess UV exposure. Even though there were some clinicopathological and ethnic dissimilarities (Table 1) and methodological differences between the two cohorts in predicting UV signature status (WGS versus WES, respectively), we observed consistent findings in both at the CpG, gene and biological pathway levels. Moreover, UV-related DNA methylation alterations showed similar distributions between BCH and TCGA in hypo- and hyper-methylated regions as well as similar enrichments in regulatory and CpG density regions, in skin disorders and in immunological pathways."

Nevertheless, to further address the reviewer's comment, we added a new approach into the manuscript by performing a DMR meta-analysis across the BCH and TCGA datasets (New Figure 4b and New Supplementary Data 19 and 20). As the results demonstrate, there are 45,915 CpGs significantly differentially methylated across the two datasets between UV-mutant and non UV-mutant cutaneous melanomas (FDR < 0.05), of which a high proportion of CpGs (equal to 24,711 CpGs or equivalent to 53.8%) have exactly the same direction of effect between BCH and TCGA (Figure 4b, Supplementary Data 19). As expected, the meta-analysis has higher power (hence, yields a larger number of significant hits) than the two separate and agnostic (cohort-specific) analyses, one being on BCH and another on TCGA, which we had reported earlier in the manuscript. However, and as described, there are some clinicopathological, ethnic and methodological differences between BCH and TCGA that need to be considered and which may yield false positive findings when the two datasets are meta-analyzed (to note as well that such false positives may be further exaggerated if we had attempted a pooled analysis, in which one dataset can over-dominate or mask the other, which is why we did not additionally perform a pooled analysis but rather a meta-

analysis in which we can better monitor the heterogeneity I^2 and the directional consistency of the signals). For this reason, we preferred to prioritize in the manuscript the approach of analyzing each of BCH and TCGA separately and agnostically, while focusing on the overlaps between the two. This approach does yield less significant hits but which are less likely to be false positives, compared to the meta-analysis approach. Importantly, both approaches convene that our top most hit, *TAPBP*, which our manuscript focuses on is robustly significant in both BCH and TCGA (New Figure 4b and Supplementary Data 7,8 19 and 20). Finally, because these two approaches can be seen as complementary, each having its own pros and cons, we decided to include the meta-analysis results in the manuscript (New Fig 4B and New Supplementary Data 19 and 20) that would complement our prioritized approach of cohort-specific analysis, hence, further reinforcing (1) the significant replication potential of the findings between the two cohorts, (2) the robustness of our major findings across two different approaches and datasets, and (3) the study design of the manuscript. For comparative purposes, we now also added New Supplementary Data 18 showing the list of CpGs from the cohort-specific analysis, so that this could be directly compared to the meta-analysis CpGs of Supplementary Data 19. We thank the reviewer for the valuable comment that has substantially strengthened our manuscript overall.

Accordingly, we added this information in the Results on page 8, lines 163-191 as such:

“We complemented the cohort-specific analyses with a DMR meta-analysis across the BCH and TCGA datasets (Figure 4b). As the results demonstrate, there are 45,915 CpGs significantly differentially methylated across the two datasets between UV-mutant and non UV-mutant cutaneous melanomas (FDR < 0.05), of which a high proportion of CpGs (equal to 24,711 CpGs or equivalent to 53.8%) have the same direction of effect between BCH and TCGA (Supplementary Data 19, Figure 4b). 121 meta-analysis CpGs (FDR < 0.05) overlapped with the 458 CpGs that are common between the BCH and TCGA cohort-specific analyses. As expected, the meta-analysis yields a larger number of significant hits (due to higher statistical power) than the cohort-specific analyses. However, the former is more prone to false positivity especially given some clinicopathological and ethnic dissimilarities (Table 1) and methodological differences between the two cohorts in predicting UV signature status (WGS versus WES, respectively). For this reason, (1) we additionally report the more stringent Bonferroni threshold, which yielded similar results as FDR (Supplementary Data 20, Figure 4b), and (2) we present the meta-analysis results as a complementary method that reinforces the robustness of the findings across the different cohorts and analysis approaches, while prioritizing the more conservative cohort-specific analysis which yields signals that are common between BCH and TCGA and which, though less profuse, are less prone to error.”

Thus, we further investigated whether the 36 CpGs in common between BCH and TCGA could be used to predict the survival of patients with cutaneous melanoma.... Notably, TAPBP differential methylation is robustly significant in both the cohort-specific and meta-analyses of the BCH and TCGA cohorts (Figure 4b and Supplementary Data 7,8 19 and 20).”

We also modified the legend of Figure 4 by adding the following:

“b) DMR fixed effects inverse variance weighted meta-analysis of BCH and TCGA and comparison between the meta-analysis and the cohort-specific analysis.”

Finally, we added the following information to the Methods on page 25, lines 539-548:

“We complemented the cohort-specific analysis by a meta-analysis across the BCH and TCGA cohorts comparing UV-mutant versus non UV-mutant cutaneous melanoma patients. We used the Metal tool⁴⁷ and the Dmrff package in R to perform DMR fixed effects inverse variance weighted meta-analysis⁴⁸, using the crude model as prioritized in the cohort-specific analysis. The meta-analysis lambda value was 1.16, showing low inflation. Statistically significant DMRs were defined as those with FDR-adjusted P value < 0.05. Due to the larger number of

hits expected with the increased statistical power afforded by the meta-analyses, we also reported the more stringent Bonferroni-adjusted p values, especially considering the higher likelihood of false positivity due to clinicopathological, ethnic and methodological differences between meta-analysed BCH and TCGA.”

8. Figure 5: Although it is interesting that TAPBP is downregulated based on increased methylation, however, it is very concerning there is no functional validation presented (i.e. disrupting the TAPBP promoter, epistasis experiments for TAPBP on proliferation, invasion, etc).

We have already addressed this comment above, in response to the reviewer’s overall assessment paragraph, where we elaborated that the mechanistic insight of this work would require a series of subsequent independent studies that build on the findings presented in the current manuscript.

9. Minor: Figure 5D is not shown.

Thank you for noting this. It was corrected accordingly to 5C.

10. Figure 6D: It is unclear if this is also based on methylation status? If not, this does not seem to be new information and should be performed based on methylation.

Figure 6d is not based on methylation status but complements Figure 6c which is. Figure 6c demonstrates that, based on the DNA methylome landscape, non UV-mutant cutaneous melanomas resembles more the pathologically different acral melanomas than the pathologically related UV-mutant cutaneous melanomas. Then Figure 6d complements these findings by showing that the non UV-mutant cutaneous melanoma patients presented worse prognosis, more closely resembling that of acral melanoma patients (known to have poorer prognosis) rather than that of UV-mutant cutaneous melanoma patients. This is also the first time that survival of acral melanoma is compared to that of UV-mutant and non UV-mutant melanoma patients.

11. Figure 6E: Were these pathways also identified in the TCGA cohort? This should be shown or referenced.

As explained in the manuscript, these pathways were identified by comparing non UV-mutant acral and cutaneous melanomas (to eliminate the effect of UV exposure) and were based on BCH data (owing to the fact that there are no acral samples in the TCGA cohort). We elaborated further on this in point 7 of Reviewer #1.

REVIEWERS' COMMENTS

Reviewer #1 (Remarks to the Author):

The authors have addressed most of my comments in the previous round of review. But I still have some questions.

Figure 2B. The authors claim that "the distribution of the detected signals was similar between the two cohorts, whether across genomic locations or in the direction of the effects (both studies had approximately 60% hypermethylation and 40% hypomethylation sites), as demonstrated in Figure 2B."

But there is no way a reader can tell the similarity from the current Fig 2B. I don't see any similarity to be honest. To show similarity, a better illustration is needed, either correlation of methylation levels, or different of methylation levels between BCH and TCGA, plotted along the genome. The two plots look not at all similar at the present form, unfortunately.

Figure 5B. there is no way that a reader can tell how the 4 types of data are integrated in this plot. Why not just display the 39 CpGs as a table? If the authors insist of keeping the circular plot, they need to explain how is the height is measured in expression, CNV, methylation and mutation, respectively. Additionally, they need to explain in the plot, how are the 4 types of data are integrated to form the driver score. As of right now, one can not get that from the plot.

Reviewer #2 (Remarks to the Author):

I am very happy with the answers the authors have provided to my points - even though these were all minor, they have addressed them in full and included clarifications in their manuscript which have made it easier to follow and understand in my opinion. I have no further comments and would support publication. I want to thank the authors for engaging in a fruitful discussion and I am looking forward to following their future projects.

Reviewer #3 (Remarks to the Author):

Thank you to the authors to attempt to address the comments. However, I believe many of my concerns during the first review still remain. While I applaud authors for the DNA methylation profiling data generated in the patients from non-european decent, which is a rarity (as pointed out), the results are correlations at best. Correlation between UV exposure and DNA methylation is very interesting, yet no mechanisms are provided. There are discrepancies between their cohort and TCGA cohort. Several results that were significant were only trends without significance in TCGA cohort. While TAPBP methylation pattern is very interesting, without thorough functional validation, the significance of this observation in melanoma biology is cannot be ascertained.

Reviewer #1 (Remarks to the Author): expertise in clinical statistics and epidemiology:

The authors have addressed most of my comments in the previous round of review. But I still have some questions.

Figure 2B. The authors claim that “the distribution of the detected signals was similar between the two cohorts, whether across genomic locations or in the direction of the effects (both studies had approximately 60% hypermethylation and 40% hypomethylation sites), as demonstrated in Figure 2B.”

But there is no way a reader can tell the similarity from the current Fig 2B. I don't see any similarity to be honest. To show similarity, a better illustration is needed, either correlation of methylation levels, or different of methylation levels between BCH and TCGA, plotted along the genome. The two plots look not at all similar at the present form, unfortunately.

Figure 2B shows the ‘overall’ chromosome distribution of methylation signals in BCH and TCGA. There are more red (hypermethylation) and blue (hypomethylation) signals in BCH, compared with TCGA, because there are more significant results in BCH as explained in the manuscript and the previously submitted revised version. However, the proportion of hypermethylation-to-hypomethylation (red-to-blue) and the distribution of signals across the chromosomes were similar between the two datasets (and quantified in the text of the manuscript as pointed out by the reviewer). To complement Figure 2B with more granular information, we had included new figures and tables in the previously submitted revised version although we had failed to bring this to the attention of the reviewer (they were rather mentioned in overall summary of the revisions provided to the editor and reviewers). These are New Supplementary Data 19-20 (added in the previous revisions): these tables show for each CpG/gene the meta-analysis results comparing BCH and TCGA, including the chromosome location, genomic coordinates and statistical similarities, differences, effect sizes, p-values, and directionality of effects (and other related information as well). This detailed information is also summarized in the new Figure 4b (added in the previous revisions). More complementary information can be found for each of BCH and TCGA in Supplementary Data 2 and 6, respectively. These tables show for each CpG/gene in each dataset the statistical effect sizes, p-values, chromosome location, genomic coordinates, and other information.

Figure 5B. there is no way that a reader can tell how the 4 types of data are integrated in this plot. Why not just display the 39 CpGs as a table? If the authors insist of keeping the circular plot, they need to explain how is the height is measured in expression, CNV, methylation and mutation, respectively. Additionally, they need to explain in the plot, how are the 4 types of data are integrated to form the driver score. As of right now, one can not get that from the plot.

We thank the reviewer for this comment. The circular plot is quantitative and provides a visual assessment of all 4 omics data types for every CpG. Adding a y-axis (height) for each OMICs layer could be useful but not easily discernable. Therefore, in this revised version, we summarized those omics heights for each CpG in the table that is linked in Figure 5b to the circular plot. The exact scores for each of the methylome (Meth), transcriptome (RNA), CNV and mutation (SNV) alterations for every CpG are now shown, as well as the overall driver score for each CpG. The figure legend has been modified accordingly, and the scoring algorithms have been further detailed in the Methods (Page 26, Lines 594-605 in the clean version of the revised manuscript).

Reviewer #2 (Remarks to the Author): expert in melanoma -omics:

I am very happy with the answers the authors have provided to my points - even though these were all minor, they have addressed them in full and included clarifications in their manuscript which have made it easier to follow and understand in my opinion. I have no further comments and would support publication. I want to thank the authors for engaging in a fruitful discussion and I am looking forward to following their future projects.

We thank the reviewer for the relevant comments that improved our manuscript.

Reviewer #3 (Remarks to the Author): expert in cancer epigenetics:

Thank you to the authors to attempt to address the comments. However, I believe many of my concerns during the first review still remain. While I applaud authors for the DNA methylation profiling data generated in the patients from non-european decent, which is a rarity (as pointed out), the results are correlations at best. Correlation between UV exposure and DNA methylation is very interesting, yet no mechanisms are provided. There are discrepancies between their cohort and TCGA cohort. Several results that were significant were only trends without

significance in TCGA cohort. While *TAPBP* methylation pattern is very interesting, without thorough functional validation, the significance of this observation in melanoma biology is cannot be ascertained.

We thank the reviewer for the comment. There are several ways to reinforce causality of associations and provide a functional assessment of the mechanisms. The reviewer had pointed out in the previous revisions one way, being “functional validation studies” and provided nice examples “Some of the key experiments would be testing how blocking DNA methylation influences *TAPBP* expression, and whether this can further influence proliferation, invasion, metastasis, etc.” While we agree that this is ‘one’ important angle, such experimental settings would require cell, organoid and/or animal models, the findings of which are not directly translatable to the human-based clinical models used in this work. This represents a major limitation of such experimental models in the context of our work. We refer the reviewer to the articles by Fedak *et al* (Fedak 2015) and Glass *et al* (Glass 2013), which cover several interesting criteria detailing how to reinforce causality of associations from various angles. Indeed, experimental manipulation represents one of nine criteria (criteria #8) (Fedak 2015). Other criteria are also important (Fedak 2015) and were applicable in the setting of this manuscript; these are: (1) “consistency”: our top findings were reproduced in two independent populations, including different ethnicities, which offer a natural means of effect randomization (hence, minimizing the influence of confounders); (2) “biological gradient”: the relation between *TAPBP* methylation (being the top hit) and patient survival showed a dose-response (*TAPBP* hypomethylation was associated with increased survival relative to hypermethylation); (3) “plausibility” and (4) “analogy”: our multi-OMICs drive score was in line with Consensus driver, which is known to be a strong indicator of cancer driver potential (Bertrand 2018); and (5) “coherence”: UV-mutant cutaneous melanoma is associated with increased patient survival, in line with a previous report (Trucco 2019), UV-mutant cutaneous melanoma is associated with *TAPBP* hypomethylation, and *TAPBP* hypomethylation in cutaneous melanoma is associated with increased patient survival. Still, more analysis and datasets will be needed to better reinforce the causality of the associations.

The differences between BCH and TCGA are expected, and it is normal to detect them, in line with the fact that BCH and TCGA represent real rather than simulated data and are based on independent populations including different ethnicities. Moreover, it would be surprising to find a complete overlap in the findings between any two datasets, especially when based on human incident cancer sample series and real population settings that involve complex biology, as compared to experimentally manipulated settings involving specific and well-defined exposure agents and doses applied to clonal (isogenic) models of cells or animals. Still, we reiterate the high level of replication we have observed between the two analyzed cohorts, including:

- 1- Similar proportions of hypermethylated (60%) and hypomethylated (40%) DMRs;
- 2- Similar enrichment in CpG regulatory regions, specifically enrichment of hypomethylated DMRs in regions 1-5Kb upstream of the transcription start site and of hypermethylated DMRs in promoters, exon/intron boundaries and 5'UTR.
- 3- Similar enrichment in CpG density regions, specifically enrichment of hyper- or hypo-methylated DMRs in CpG islands or shores.
- 4- A large portion of epigenetically deregulated pathways in common (10 identical pathways out of 28 and 30 pathways in BCH and TCGA, respectively);
- 5- Reproducible effects of *TAPBP* methylation on patient prognosis;
- 6- Statistically significant enrichment of findings between the two cohorts, even when zoomed to single CpG resolution;
- 7- Validation of the top marker (*TAPBP* methylation) by an independent technique (pyrosequencing).
- 8- Reinforcement of the observed enrichment of findings between BCH and TCGA datasets by the meta-analysis results across the two cohorts, which was performed following the reviewer’s comments in the previous revision (Supplementary Data 19-20 and Figure 4b, added in the previous revisions).

References:

Bertrand D, Drissler S, Chia BK, Koh JY, Li C, Suphavitai C, Tan IB, Nagarajan N. ConsensusDriver Improves upon Individual Algorithms for Predicting Driver Alterations in Different Cancer Types and Individual Patients. *Cancer Res.* 2018 Jan 1;78(1):290-301. doi: 10.1158/0008-5472.CAN-17-1345. Epub 2017 Dec 19. PMID: 29259006.

Fedak KM, Bernal A, Capshaw ZA, Gross S. Applying the Bradford Hill criteria in the 21st century: how data integration has changed causal inference in molecular epidemiology. *Emerg Themes Epidemiol.* 2015 Sep 30;12:14. doi: 10.1186/s12982-015-0037-4. PMID: 26425136; PMCID: PMC4589117.

Glass TA, Goodman SN, Hernán MA, Samet JM. Causal inference in public health. *Annu Rev Public Health.* 2013;34:61-75. doi: 10.1146/annurev-publhealth-031811-124606. Epub 2013 Jan 7. PMID: 23297653; PMCID: PMC4079266.

Trucco LD, Mundra PA, Hogan K, Garcia-Martinez P, Viros A, Mandal AK, Macagno N, Gaudy-Marqueste C, Allan D, Baenke F, Cook M, McManus C, Sanchez-Laorden B, Dhomen N, Marais R. Ultraviolet radiation-induced DNA damage is prognostic for outcome in melanoma. *Nat Med.* 2019 Feb;25(2):221-224. doi: 10.1038/s41591-018-0265-6. Epub 2018 Dec 3. Erratum in: *Nat Med.* 2018 Dec 18;: PMID: 30510256.